

# A land surface model combined with a crop growth model for paddy rice (MATCRO-Rice Ver. 1) – Part I: Model description

Yuji Masutomi[1], Keisuke Ono[2], Masayoshi Mano[3], Atsushi Maruyama[4], and Akira Miyata[2]

[1]College of Agriculture, Ibaraki University, 3-21-1, Chuo, Ami, Inashiki, Ibaraki 300-0393, Japan
[2]National Institute for Agro-Environment Sciences, 3-1-3, Kannondai, Tsukuba, Ibaraki 305-8604, Japan
[3]Graduate School of Horticulture, Chiba University, 648 Matsudo, Matsudo-shi, Chiba 271-8510, Japan
[4]National Agriculture and Food Research Organization, 3-1-1 Kannondai, Tsukuba, Ibaraki 305-8666, Japan

*Correspondence to:* Yuji Masutomi (yuji.masutomi@gmail.com)

**Abstract.** Crop growth and agricultural management can affect climate at various spatial and temporal scales through the exchange of heat, water, and gases between land and atmosphere. Therefore, accurate simulation of fluxes for heat, water, and gases from agricultural land is important for climate simulations. A land surface model (LSM) combined with a crop growth model (CGM), called LSM-CGM combined model, is a useful tool for simulating these fluxes from agricultural land. Therefore, we developed a new LSM-CGM combined model for paddy rice fields, the MATCRO-Rice model. The main objective of this paper is to present the full description of MATCRO-Rice. The most important feature of MATCRO-Rice is that it can consistently simulate latent and sensible heat fluxes, net carbon flux, and crop yield by exchanging variables between the LSM and CGM. This feature enables us to apply the model to a wide range of integrated issues.

## 1 Introduction

In the last 15 years, climate and land surface modelling studies have shown that crop growth and farm management in agricultural land significantly affect climate via the exchange of heat, water, and gases. For example, applying a regional climate model combined with a crop growth model (CGM) to the United States, Tsvetsinskaya et al. (2001) showed that crop growth can change the surface temperature by 2 to 4°C. Maruyama and Kuwagata (2010) showed that crop growing season can affect the amount of evapotranspiration by using a land surface model (LSM) combined with a CGM. Levis et al. (2012) incorporated a CGM into an earth system model, and showed that the timing of crop sowing can change the amount of precipitation. Using a dynamic global vegetation model combined with a CGM, Bondeau et al. (2007) showed that the global carbon cycle, which has a significant effect on global warming, is largely modified by crop growth and farm management. Osborne et al. (2009), using a global climate model coupled with a CGM, demonstrated that the crop–climate interaction can affect annual variability in surface temperature. All these studies indicate that crop growth and farm management are key determinants of climate and that climate simulations need to accurately simulate the fluxes of heat, water, and gases in agricultural land.

A LSM or dynamic vegetation model (DVM) incorporated with a CGM, called LSM-CGM or DVM-CGM combined models, are a useful tool for simulating the fluxes of heat, water, and gases in agricultural land. Hence, several LSMs and DVMs incorporated with a CGM have been developed (BATS-GF: Tsvetsinskaya et al., 2001; Agro-IBIS: Kucharik, 2003;





ORCHIDEE-STICS: Gervois et al., 2004; LPJmL: Bondeau et al., 2007; GLAM-MOSES2: Osborne et al., 2007; SIBcrop: Lokupitiya et al., 2009; MK10: Maruyama and Kuwagata, 2010; CLM4CNcrop: Levis et al., 2012; JULES-crop: Osborne et al., 2015). Lei et al. (2010) divided these incorporated models into three types in terms of integration schemes for the leaf area index (LAI). Among these types, the type of models that consistently simulate crop production, LAI, water-energy flux, and

5 carbon flux by exchanging variables between an LSM and a CGM allows for wide applicability and comprehensive evaluation of the model with observations (Lei et al., 2010). However, this type comprises currently only four models: Agro-IBIS, SIBcrop, CLM4CNcrop, and JULES-crop. Among these, only JULES-crop can simulate the growth of rice, although rice is one of the major crops, accounting for 23% of agricultural land farmed with cereals worldwide (FAO, 2015). Nevertheless, the JULES-crop model does not consider a flooded surface of paddy rice fields, which is an important parameter when simulating

heat and water fluxes in paddy rice fields, because heat and water fluxes in a flooded surface are largely different from those in a non-flooded surface.

We developed a new LSM-CGM model, called MATCRO-Rice. The aim of this paper is to describe the MATCRO-Rice model in detail. The most important feature of MATCRO-Rice is that it can consistently simulate latent heat flux (LHF), sensible heat flux (SHF), net carbon flux, and crop yields by exchanging variables between LSM and CGM. Herein, we first

provide the overview of MATCRO-Rice in Section 2, and then describe the LSM and CGM of MATCRO-Rice in detail in Sections 3 and 4, respectively. Last, we discuss the applications and limitations of MATCRO-Rice in Section 5. The model validation for MATCRO-Rice is described in the accompanied paper (Masutomi et al., 2016).

## 2 Model overview: MATCRO-Rice

MATCRO-Rice has two main components: LSM and CGM. The LSM component mainly simulates LHF and SHF. It is based

on MATSIRO (Takata et al., 2003), which is embedded in global climate models (MIROC5.0: Watanabe et al., 2010; NICAM: Satoh et al., 2008) and a climate system model (MIROC-ESM: Watanabe et al., 2011). In addition, MATSIRO is used for a range of hydrological applications (e.g., Pokhrel et al., 2012; Hirabayashi et al., 2013).

The CGM of MATCRO-Rice mainly simulates rice yield and biomass for each organ during a growing period. The CGM used in MATCRO-Rice is based on CGMs developed by the School of de Wit (Bouman et al., 1996; e.g., MACROS: Penning

de Vries et al., 1989; SUCROS: Goudriaan and van Laar, 1994; ORYZA2000: Bouman et al., 2001).

The meteorological inputs to run MATCRO-Rice are listed in Table 1. The standard outputs of MATCRO-Rice are LHF, SHF, biomass of organs during a growing period, and crop yield. All other variables simulated in MATCRO-Rice can be output if needed. The feature of MATCRO-Rice is to exchange variables between the LSM and CGM. The variables exchanged are listed in Table 2.





## 3 Land surface model

The main outputs of the LSM of MATCRO-Rice are LHF and SHF. The LSM has five modules, which are "energy balance at the canopy and surface water", "within-canopy shortwave radiation", "bulk transfer coefficient for latent and sensible heat", "canopy water balance", and "soil water and heat transfer". Each module is described in detail in the following sections. Before describing each module, we note the following two major modifications from the original LSM, MATSIRO (Takata et al., 2003).

1. LAI, crop height, and root depth, which are constant in the original MATSIRO, are dynamically calculated in the CGM and are the inputs to the LSM.

2. Surface water is added above the soil surface to represent a flooded surface in paddy rice fields.

Other minor modifications are described separately in each of the following sections. We note that the photosynthesis model used in MATCRO is described in the CGM section (Section 4).

### 3.1 Energy balance at the canopy and surface water

This module calculates LHF and SHF by solving energy balance at two layers above the soil, canopy and surface water. The module is based on the original MATSIRO (Takata et al., 2003), except for the addition of surface water above the soil and other minor modifications. The energy balance at the canopy and surface water are given as follows:

$$R_{nc} = H_c + \lambda E_c + \lambda E_t, \qquad \text{(Canopy)} \qquad (1)$$

$$R_{nw} = H_w + \lambda E_w + G_{ws} + S_{tw}, \qquad \text{(Water surface)} \qquad (2)$$

where $R_{nc}$ and $R_{nw}$ are the net radiant flux density at canopy and surface water, $H_c$ and $H_w$ are the SHF from the canopy and surface water, $E_c$, $E_t$, and $E_w$ are the evaporation from wet canopy, transpiration from the canopy, and evaporation from the surface water, respectively, $G_{ws}$ is the heat flux from the surface water to soil, and $S_{tw}$ is the heat flux stored into surface water. It is important to note that the downward flux for $R_{nc}$, $R_{nw}$, and $G_{ws}$ indicates a positive flux, whereas downward flux for $H_c$, $H_w$, $E_c$, $E_t$, and $E_w$ indicates a negative flux. All variables in the model are listed in Table 3. $\lambda$ is the physical constant for the



latent heat of vaporisation (Table 4). Each radiant, heat, and water flux in Eqs. 1 and 2 are given by the following equations.

$$R_{nc} = (R_s^d(0) - R_s^u(0))(1 - \tau_{cs}) + \epsilon R_l^d(0)(1 - \tau_{cl}) - (2\epsilon\sigma T_c^4 - \epsilon\sigma T_w^4)(1 - \tau_{cl}), \tag{3}$$

$$R_{nw} = (R_s^d(0) - R_s^u(0))\tau_{cs} + \epsilon R_l^d(0)\tau_{cl} - \epsilon\sigma T_w^4 + \epsilon\sigma(1 - \tau_{cl})T_c^4, \tag{4}$$

$$H_c = c_{pa}\rho_a C_{Hc} U(T_c - T_a), \tag{5}$$

$$H_w = c_{pa}\rho_a C_{Hw} U(T_w - T_a), \tag{6}$$

$$E_c = f_{cw}\rho_a C_{Hc} U(Q_{sat}(T_c, P_a) - Q), \tag{7}$$

$$E_t = (1 - f_{cw})\rho_a C_{Ec} U(Q_{sat}(T_c, P_a) - Q), \tag{8}$$

$$E_w = \rho_a C_{Ew} U(Q_{sat}(T_w, P_a) - Q), \tag{9}$$

$$G_{ws} = k_w(T_w - T_s(0))/d_w, \tag{10}$$

$$S_{tw} = c_{pw}\rho_w d_w(dT_w/dt), \tag{11}$$

where $R_s^d(0)$, $R_l^d(0)$, and $R_s^u(0)$ are the downward shortwave radiant flux density, downward longwave radiant flux density, and upward shortwave radiant flux density at the canopy top, respectively, $\tau_{cs}$ and $\tau_{cl}$ are the canopy transmissivity for shortwave and longwave radiation, respectively, $C_{Hc}$ and $C_{Hw}$ are the bulk transfer coefficients (BTCs) for sensible heat between canopy and atmosphere and between surface water and atmosphere, respectively, $C_{Ec}$ and $C_{Ew}$ are the BTCs for latent heat between canopy and atmosphere and between canopy and atmosphere, respectively, $T_a$, $P_a$, $U$, and $Q$ are air temperature, air pressure, wind speed, and specific humidity, respectively, $f_{cw}$ is the fraction of wet canopy, $T_c$, $T_w$, and $T_s(0)$ are the canopy, surface water, and soil surface temperature, respectively, $c_{pa}$ and $c_{pw}$ are the specific air and water heat, respectively, $k_w$ is the water thermal conductivity, $\rho_w$ and $\rho_a$ are water and air density, respectively, $\sigma$ is the Boltzmann constant, $Q_{sat}$ is specific humidity at saturation, $d_w$ is the depth of surface water, $\epsilon$ is the longwave emissivity of surface water, and $d/dt$ indicates the time differentiation. The argument of the radiant flux density denotes LAI depth from the canopy top, and the argument of soil temperature denotes soil depth from the soil surface. Therefore, $R_s^d(0)$, $R_l^d(0)$, and $R_s^u(0)$ indicate the radiant flux density at the canopy top, and $T_s(0)$ indicates the soil surface temperature.

$T_a$, $P_a$, $U$, $Q$, $R_s^d(0)$, and $R_l^d(0)$ are meteorological forcing inputs (Table 1). $R_s^u(0)$, $\tau_{cs}$, $\tau_{cl}$, $f_{cw}$, $C_{Ec}$, $C_{Ew}$, $C_{Hc}$, $C_{Hw}$, and $T_s(0)$ are calculated from Eqs. 21, 20, 23, 39, 25, 24, 27, 26, and 45, respectively, which are given in the following sections. The variables $\rho_a$ and $Q_{sat}$ are physically calculated from the air temperature and air pressure (Appendix A), $c_{pa}$, $c_{pw}$, $k_w$, $\rho_w$, and $\sigma$ are physical constants (Table 4), $d_w$ is a simulation setting parameter (Table 5), and $\epsilon$ is set to 0.96 (Campbell and Norman, 1998). $T_c$ and $T_w$ are numerically determined to satisfy Eqs. 1 to 11. The numerical method is described in Masutomi et al. (2016).

The original MATSIRO uses $C_{Hc}$ instead of $C_{Ec}$ in Eq. 8 when specific humidity of the air is greater than the saturated specific humidity of the canopy (i.e., $Q_{sat} - Q < 0$), because dew condensation occurs at canopy of interest. MATCRO does not consider the effect for simplicity. It should be noted that $C_{Hc}$ is used for calculating the evaporation from wet canopy in Eq. 7.



## 3.2 Within-canopy shortwave radiation

The main role of this module is to simulate direct downward photosynthesis active radiation (PAR), scattered downward PAR, and scattered upward PAR at a LAI depth of $l$ from the canopy top by calculating the transmission and reflection of shortwave radiation by leaves within canopies. These PARs are used for calculating carbon assimilation in the CGM (Section 4.1). In addition to the simulation of PARs, transmissivities for shortwave and longwave radiation are simulated in this module. The transmissivities are used for calculating LHF and SHF (Section 3.1).

This module is based on the simple model developed by Watanabe and Ohtani (1995). The model determines radiation within canopies by calculating the transmission and reflection of the radiation within the canopy. In this model, radiation within the canopy is divided into three components (downward direct, downward scattered, and upward scattered) and two wavebands (PAR and near infrared [NIR]). In addition, the following three assumptions are considered in the model for simplicity.

1. Leaf orientation is random (i.e., spherical distribution).

2. Leaf reflectivity and transmissivity of the radiation are vertically uniform within a canopy.

3. Scattered radiation income from a zenith angle of $53°$.

It should be noted that the assumption 3 is based on the fact that radiant flux uniformly emitted from a horizontal plane is approximately equal to radiant flux density from a zenith angle of $53°$. From the three assumptions above, we can express analytically the radiant flux density for downward direct ($D_i^d(l)$), downward scattered ($S_i^d(l)$), and upward scattered ($S_i^u(l)$) within canopy for each waveband ($i = 1$: PAR; $i = 2$: NIR), as follows:

$$D_i^d(l) = D_i^d(0)\exp(-Fl\sec(\theta)), \tag{12}$$

$$S_i^d(l) = C_{1,i}\exp(a_i l) + C_{2,i}\exp(-a_i l) + C_{3,i}D_i^d(l), \tag{13}$$

$$S_i^u(l) = A_{1,i}C_{1,i}\exp(a_i l) + A_{2,i}C_{2,i}\exp(-a_i l) + C_{4,i}D_i^d(l). \tag{14}$$

Here, $F$ is a parameter for the distribution of leaf orientation. If we assume spherical distribution for leaf orientation as mentioned above, we have $F = 0.5$ (Goudriaan and van Laar (1994)). The variable $l$ is a LAI depth from the canopy top. The variable $\theta$ is a zenith angle of the sun (Appendix B). The function $\sec()$ indicates the secant function. The coefficients, $a_i$, $C_{1,i}$, $C_{2,i}$, $C_{3,i}$, $C_{4,i}$, $A_{1,i}$, and $A_{2,i}$ are calculated as shown in Appendix C. It should be noted that $a_i$ indicates the extinction coefficient for scattered radiation. $D_i^d(0)$ is obtained by splitting radiant flux density for downward shortwave at the top of the canopy into direct and scattered radiation as follows:

$$D_i^d(0) = 0.5R_s^d(0)(1 - f_{df}), \tag{15}$$

$$S_i^d(0) = 0.5R_s^d(0)f_{df}, \tag{16}$$

where $R_s^d(0)$ is the downward shortwave radiant flux density at the canopy top and $f_{df}$ is the fraction of scattered radiation to total radiation. In Eqs. 15 and 16, we assumed that both PAR and NIR are half of $R_s^d(0)$. According to Goudriaan and van Laar



(1994), $f_{df}$ is given as a function of the transmissivity of atmosphere ($\tau_{atm}$) as follows:

$$
f_{df} = \begin{cases} 1 & (\tau_{atm} < 0.22) \\ 1 - 6.4(\tau_{atm} - 0.22)^2 & (0.22 \leq \tau_{atm} < 0.35), \\ 1.47 - 1.66\tau_{atm} & (\text{Otherwise}) \end{cases} \tag{17}
$$

$$
\tau_{atm} = R_s^d(0)\sec(\theta)/R_{ex}, \tag{18}
$$

$$
R_{ex} = R_{sun}(1 + 0.033)\cos(2\pi(D_{oy}/365)), \tag{19}
$$

where $R_{ex}$ is the extraterrestrial radiation, $R_{sun}$ is the solar constant, and $D_{oy}$ is the number of days from Jan 1. The equations 15–19 that calculate $D_i^d(0)$ are based on formulations by Goudriaan and van Laar (1994), while the original MATSIRO uses different equations.

The transmissivity of canopies for shortwave radiation ($\tau_{cs}$) is expressed as

$$
\tau_{cs} = R_s^d(L)/(R_s^d(0) - R_s^u(0)). \tag{20}
$$

Here, $R_s^u(0)$ and $R_s^d(L)$ are the radiant flux density for upward shortwave at the canopy top and downward shortwave at the bottom of the canopy, respectively. $L$ denotes the LAI, which is calculated in the CGM (Section 4.4). $R_s^u(0)$ and $R_s^d(L)$ are represented by

$$
R_s^u(0) = r_{11}D_1^d(0) + r_{21}D_2^d(0) + r_{12}S_1^d(0) + r_{22}S_2^d(0), \tag{21}
$$

$$
R_s^d(L) = \tau_{11}D_1^d(0) + \tau_{21}D_2^d(0) + \tau_{12}S_1^d(0) + \tau_{22}S_2^d(0), \tag{22}
$$

where $r_{ij}$ and $\tau_{ij}$ are the canopy reflectivity and transmissivity, respectively, $i$ and $j$ represent wavebands ($i = 1$: PAR; $i = 2$: NIR) and direct ($j = 1$) or scattered radiation ($j = 2$). These are given in Appendix D.

Last, the transmissivity of a canopy for longwave radiation ($\tau_{cl}$) is expressed as

$$
\tau_{cl} = \exp(-FLd_f), \tag{23}
$$

where, $d_f$ is the scattered factor. We set $d_f = \sec(2\pi(53/360))$ from the assumption that scattered radiation income is from a zenith angle of $53°$ (Watanabe, 1994).

### 3.3 Bulk transfer coefficient for latent and sensible heat

This module calculates BTCs for latent and sensible heat ($C_{Ec}$, $C_{Ew}$, $C_{Hc}$, and $C_{Hw}$). The BTCs are used to simulate energy balance (Section 3.1). This module is based on Watanabe (1994), where $C_{Ew}$, $C_{Ec}$, $C_{Hw}$, and $C_{Hc}$ are given by

$$
C_{Ew} = \kappa^2 \left[\ln\left(\frac{z_a - d}{z_{Mw}}\right) + \Psi_M(\zeta_w)\right]^{-1} \left[\ln\left(\frac{z_a - d}{z_{Qw}}\right) + \Psi_E(\zeta_w)\right]^{-1}, \tag{24}
$$

$$
C_{Ec} = C_E - C_{Ew}, \tag{25}
$$

$$
C_{Hw} = \kappa^2 \left[\ln\left(\frac{z_a - d}{z_{Mw}}\right) + \Psi_M(\zeta_w)\right]^{-1} \left[\ln\left(\frac{z_a - d}{z_{Tw}}\right) + \Psi_H(\zeta_w)\right]^{-1}, \tag{26}
$$

$$
C_{Hc} = C_H - C_{Hw}, \tag{27}
$$



where $C_E$ and $C_H$ are the BTCs for latent and sensible heat between the entire surface (canopy + surface water) and atmosphere and are given by

$$C_E = \kappa^2 \left[ \ln\left(\frac{z_a - d}{z_M}\right) + \Psi_M(\zeta) \right]^{-1} \left[ \ln\left(\frac{z_a - d}{z_Q}\right) + \Psi_E(\zeta) \right]^{-1}, \tag{28}$$

$$C_H = \kappa^2 \left[ \ln\left(\frac{z_a - d}{z_M}\right) + \Psi_M(\zeta) \right]^{-1} \left[ \ln\left(\frac{z_a - d}{z_T}\right) + \Psi_H(\zeta) \right]^{-1}. \tag{29}$$

In Eqs. 24 to 29, $\kappa$ is the Karman constant, $d$ is the zero-plane displacement height, $z_a$ is the reference height at which wind velocity is observed, $z_{Mw}$, $z_{Tw}$, $z_{Qw}$ are the roughness lengths that express the effect of surface water on the profiles of momentum, temperature, and specific humidity, respectively, $z_M$, $z_T$, and $z_Q$ are the roughness lengths of an entire surface (canopy + surface water) for the profiles of momentum, temperature, and specific humidity, respectively. $z_a$ is a simulation setting parameter (Table 5), and $d$, $z_M$, $z_T$, $z_Q$, $z_{Mw}$, $z_{Tw}$, and $z_{Qw}$ are the functions of crop height and LAI (Appendix E).

$\Psi_M$, $\Psi_H$, and $\Psi_E$ are the diabatic correction factors for momentum, heat, and vapour transport, respectively. The factors are functions of atmospheric stability $\zeta$ as follows:

$$\Psi_M(\zeta) = \begin{cases} 6\ln(1+\zeta) & (\zeta > 0 : \text{stable}) \\ -1.2\ln\left[\frac{1+(1-16\zeta)^{1/2}}{2}\right] & (\text{Otherwise: unstable}), \end{cases} \tag{30}$$

$$\Psi_H(\zeta) = \Psi_E(\zeta) = \begin{cases} 6\ln(1+\zeta) & (\zeta > 0 : \text{stable}) \\ -2\ln\left[\frac{1+(1-16\zeta)^{1/2}}{2}\right] & (\text{Otherwise: stable}). \end{cases} \tag{31}$$

The equations above are adopted from Campbell and Norman (1998), whereas the original MATSRIO model employs different equations. The variable $\zeta$ is replaced by either the atmospheric stability between the entire surface and atmosphere ($\zeta$) or the atmospheric stability between surface water and atmosphere ($\zeta_w$). These are given by

$$\zeta = \frac{z_a - d}{L_{MO}}, \tag{32}$$

$$\zeta_w = \frac{z_a - d}{L_{MOw}}, \tag{33}$$

where $L_{MO}$ and $L_{MOw}$ are the Monin-Obukhov lengths for the exchange between the entire surface and atmosphere and between the surface water and atmosphere, respectively, and are given by

$$L_{MO} = \frac{\Theta_0 C_M^{3/2} U^2}{\kappa g \{ C_{Hw}(T_w - T_a) + C_{Hc}(T_c - T_a) \}}, \tag{34}$$

$$L_{MOw} = \frac{\Theta_0 C_{Mw}^{3/2} U^2}{\kappa g C_{Hw}(T_w - T_a)}, \tag{35}$$

where $g$ is the gravitational constant, $T_w$ and $T_c$ are the temperatures of the surface water and canopy, $\Theta_0$ is the potential

temperature, $C_M$ and $C_{Mw}$ are the BTC for momentum between an entire surface and atmosphere and between water surface and atmosphere, respectively. $C_{Mw}$ in Eq. 35 is introduced according to Maruyama and Kuwagata (2008), while the original





MATSIRO uses $C_M$. $T_w$ and $T_c$ are calculated in Section 3.1. $\Theta_0$ is given by

$$\Theta_0 = T_a * (1.0 * 10^5/P_a)^{(R_{dry}/c_{pa})}, \tag{36}$$

where $R_{dry}$ is the gas constant of dry air. Although the original MATSIRO fixes $\Theta_0$ at 300 K, MATCRO calculates the value according to Campbell and Norman (1998). $C_M$ and $C_{Mw}$ are given by

$$C_M = k^2 \left[\ln\left(\frac{z_a - d}{z_M}\right) + \Psi_M(\zeta)\right]^{-2}, \tag{37}$$

$$C_{Mw} = k^2 \left[\ln\left(\frac{z_a - d}{z_{Mw}}\right) + \Psi_M(\zeta_w)\right]^{-2}. \tag{38}$$

Now we have six independent equations, Eqs. 24, 25, 26, 27, 37, and 38, for six unknown variables, $C_{Ew}$, $C_{Ec}$, $C_{Hw}$, $C_{Hc}$, $C_M$, and $C_{Mw}$, respectively. Therefore, we can determine the values of these variables by numerically solving Eqs. 24 to 38. The numerical method is described in Masutomi et al. (2016).

## 3.4 Canopy water balance

The main purpose of this module is to calculate the fraction of wet canopy ($f_{cw}$) which is used for simulating energy balance at canopy (Section 3.1). To calculate $f_{cw}$, this module calculates water balance at canopy. Although the module is based on the original MATSIRO, the amount of water that canopies can hold was replaced by using the method described in Penning de Vries et al. (1989). The variable $f_{cw}$ is given as

$$f_{cw} = w_c/w_{cap}, \tag{39}$$

where $w_c$ is the amount of water stored in canopy and $w_{cap}$ is the water capacity of the canopy. The $w_c$ is calculated by solving the canopy water balance, which is given by

$$\rho_w \frac{dw_c}{dt} = I_c - D_g - E_c, \tag{40}$$

where $\rho_w$ is the density of water, $I_c$ is the amount of precipitation intercepted by canopy, $D_g$ is the amount of water that falls from the canopy onto surface water due to gravity, and $E_c$ is the amount of water that evaporates from the canopy (Eq. 7). $I_c$ depends on the amount of precipitation ($P_r$) and LAI ($L$) and is given by

$$I_c = f_{int}P_r, \tag{41}$$

$$f_{int} = \begin{cases} L & (L < 1) \\ 1 & (\text{otherwise}) \end{cases}, \tag{42}$$

where $f_{int}$ indicates the interception efficiency of precipitation by canopy. According to Rutter et al. (1975) and Penning de Vries et al. (1989), $D_g$ and $w_{cap}$ are given as

$$D_g = \rho_w D_1 \exp(D_2 w_c), \tag{43}$$

$$w_{cap} = (W_{sh} * 10^{-4})/\rho_w, \tag{44}$$




respectively, where $D_1$ and $D_2$ are parameters (Rutter et al., 1975), and $W_{sh}$ is the shoot dry weight, which is calculated in the CGM (Eq. 127).

### 3.5 Soil water and heat transfer

This module calculates heat and water transfer in soil. The main role of this module is to determine the temperature at a soil surface ($T_s(0)$), which is used for simulating energy balance of the surface water (Section 3.1). Although this module is based on the original MATSIRO, the calculations of the surface and base runoffs are simplified because hydrological calculations are not the main purpose of MATCRO-Rice.

Soil temperature at a soil depth of $z$ from the soil surface ($T_s(z)$) is calculated from the gradient of heat flux in the soil as follows:

$$c_{hs}(z)\frac{\partial T_s(z)}{\partial t} = \frac{\partial G_s(z)}{\partial z}, \tag{45}$$

where $c_{hs}$ is the volumetric heat capacity of the soil and $G_s(z)$ is the heat flux at a soil depth of $z$ and is given from the gradient of soil temperature

$$G_s(z) = \begin{cases} k_{ts}(z)\frac{\partial T_s(z)}{\partial z} & (0 \leq z < z_{max}) \\ 0 & (z = z_{max}). \end{cases} \tag{46}$$

Here, $k_{ts}$ is the soil thermal conductivity. In Eq. 46, we assumed that heat flux at the bottom of the soil layer ($z = z_{max}$) is zero. $z_{max}$ is a simulation setting parameter. When solving Eqs. 45 and 46, the heat flux from surface water to soil ($G_{ws}$), calculated in Eq. 10, is used as a boundary condition. The parameter $c_{hs}$ is calculated from the heat capacities of soil components as follows.

$$c_{hs}(z) = \rho_s c_{pm} + \rho_w c_{pw} w_s(z), \tag{47}$$

where $\rho_s$ is the bulk density of soil, $c_{pm}$ is the specific heat of soil minerals, and $w_s(z)$ is the volumetric concentration of soil water. $\rho_s$ is a soil-type specific parameter determined by soil type at a simulation site, and $c_{pm}$ is given according to Campbell and Norman (1998) . We note that the first term of the right hand side in Eq. 47 indicates the heat capacity of dry soil. Although the original MATSRIO model assigns a default value to the heat capacity of dry soil for all soil types, MATCRO-Rice calculates the value of the heat capacity of dry soil using the bulk density of soil and the heat capacity of soil minerals, as shown in the first term of Eq. 47. It should be noted that the effect of soil organic matter on $c_{hs}$ is not considered in MATCRO. The parameter $k_{ts}(z)$ in Eq. 46 is given by

$$k_{ts}(z) = K_e(z)(k_{tss} - k_{ts0}) + k_{ts0}, \tag{48}$$

$$K_e(z) = \begin{cases} \log(w_s(z)/w_{sat}) + 1.0 & (\text{if } w_s(z)/w_{sat} \geq 0), \\ 0 & (\text{otherwise}) \end{cases} \tag{49}$$

where $k_{ts0}$ and $k_{tss}$ are the thermal conductivity of dry and saturated soils, respectively, $K_e$ is the Kersten number, and $w_{sat}$ is the volumetric soil water concentration at saturation. $k_{ts0}$ and $k_{tss}$ are parameters. We set $k_{ts0}$=0.25 (Campbell and Norman,



1998), and $k_{tss}$ = 1.58 (Best et al., 2011). The parameter $w_{sat}$ is specific to soil type. Equations 48 and 49 for the calculation of $k_{ts}(z)$ are based on the equations developed by Best et al. (2011), while the original MATSIRO employs a different equation. The variable $w_s(z)$ depends on the gradient of water flux and absorption by roots at a soil depth $z$ and is given by

$$
\begin{aligned}
w_s(z) &= w_{sat} & (0 \leq z \leq z_{sat}), \\
\frac{\partial w_s(z)}{\partial t} &= \frac{\partial F_s(z)}{\partial z} + S_s(z) & (z_{sat} < z \leq z_{max}),
\end{aligned}
$$
(50)
(51)

where $F_s(z)$ and $S_s(z)$ are water flux and absorption by roots at a soil depth of $z$, respectively. For simplicity, the top soil layer is assumed to be saturated, because the surface above soil is flooded. Given the assumption, we do not need to explicitly simulate water flow from a flooded surface into soil. This assumption is not considered in the original MATSIRO. $z_{sat}$ is a simulation setting parameter. $F_s(z)$ is calculated from the gradient of water potentials as follows.

$$
F_s(z) = \begin{cases} -K(z)\left(\frac{\partial \psi(z)}{\partial z} + 1\right) & (0 \leq z \leq z_b) \\ (w_{sat}/\tau_b)(w_s(z)/w_{sat})^2 & (z_b < z \leq z_{max}) \end{cases},
$$
(52)

where $K(z)$ is the hydraulic conductivity and $\psi(z)$ is the water potential at a soil depth of $z$. $F_s(z)$ in the bottommost layer ($z_b < z < z_{max}$) represents the base flow, and $\tau_b$ is the recession constant for base flow. This model uses a simple model for simulating base flow developed by Hanasaki et al. (2008), although the original MATSIRO utilizes a more complicated model (TOPMODEL: Beven and Kirkby (1979)). $z_b$ is a simulation setting parameter, and $\tau_b$ is determined as described in Hanasaki et al. (2008). $K(z)$ and $\psi(z)$ are given by Clapp and Hornberger (1978) as follows.

$$
\begin{aligned}
K(z) &= K_s \left(\frac{w_s(z)}{w_{sat}}\right)^{2B+3}, \\
\psi(z) &= \psi_s \left(\frac{w_s(z)}{w_{sat}}\right)^{-B},
\end{aligned}
$$
(53)
(54)

where $K_s$ and $\psi_s$ are hydraulic conductivity and water potentials at saturation, respectively, and $B$ is a parameter that determines the relationship of hydraulic conductivity or water potentials between saturated and unsaturated soils. $K_s$, $\psi_s$, and $B$ are soil-type specific parameters. $S_s(z)$ in Eq. 51 is calculated from the transpiration

$$
S_s(z) = \begin{cases} E_t/(\rho_w z_{rt}) & (0 \leq z \leq z_{rt}) \\ 0 & (z_{rt} < z \leq z_{max}) \end{cases},
$$
(55)

where $E_t$ is the transpiration calculated in Eq. 8 and $z_{rt}$ is a root depth calculated by the CGM (Eq. 131). In Eq. 55, we assumed that $S_s(z)$ has no dependency on soil depth.

## 4 Crop growth model

The main purpose of the CGM is to simulate rice yield and biomass growth for each organ during a growing period. The CGM has four modules: "net carbon assimilation", "crop development", "crop growth", and "LAI, height, and root depth". Each module is described in detail in the following sections.





## 4.1 Net carbon assimilation

The main role of this module is to calculate net carbon assimilation ($A_n$) in canopy for simulating crop growth. In addition, the stomatal conductance per unit leaf area for both sides of the leave ($\overline{g}_s$) is calculated for simulating roughness length (Appendix E). Although this module is based on the Big-leaf model (Sellers et al., 1992, 1996a) used in the original MATSIRO, we refined two points in the calculation according to the approach described by de Pury and Farquhar (1997) and Dai et al. (2004). The first refinement is that leaves in a canopy are divided into sunlit and shade leaves. Subsequently, $A_n$ per unit leaf area for each the sunlit and shade leaves are calculated. The second refinement is that $A_n$ for the entire canopy is calculated considering vertical distribution of nitrogen within the canopy.

$A_n$ for the entire canopy is given by

$$A_n = \overline{A}_{n,sn}L_{sn} + \overline{A}_{n,sh}L_{sh}, \tag{56}$$

where $\overline{A}_{n,sn}$ and $\overline{A}_{n,sh}$ are net carbon assimilation per unit leaf area for sunlit and shade leaves, respectively, $L_{sn}$ and $L_{sh}$ are LAI for sunlit and shade leaves, respectively, and overbars represent the amounts per unit leaf area. $\overline{A}_{n,sn}$ and $\overline{A}_{n,sh}$ are defined by the difference between gross carbon assimilation and respiration as follows:

$$\overline{A}_{n,x} = \overline{A}_{g,x} - \overline{R}_{d,x}, \tag{57}$$

where $\overline{A}_{g,x}$ and $\overline{R}_{d,x}$ are gross carbon assimilation and respiration per unit leaf area, respectively, and the suffix $x$ indicates $sn$ or $sh$. $L_{sn}$ and $L_{sh}$ are given as follows.

$$L_{sn} = \int_0^L f_{sn}(l)dl, \tag{58}$$

$$L_{sh} = \int_0^L (1 - f_{sn}(l))dl, \tag{59}$$

where $f_{sn}(l)$ is the fraction of sunlit leaves at a LAI depth of $l$ and is defined as follows:

$$f_{sn}(l) = \exp(-Fl\sec(\theta)), \tag{60}$$

where $F$ denotes distribution of leaf orientation and $\theta$ is a zenith angle of the sun (Appendix B). The effect of photosynthesis down-regulation due to acclimatization to elevated $CO_2$ is represented as follows:

$$\overline{A}_{g,x} = f_{dwn} * \overline{A}_{g',x}, \tag{61}$$

$$f_{dwn} = \{1 + \gamma_{gd}\ln(C_a/C_0)\}/\{1 + \gamma_g\ln(C_a/C_0)\}, \tag{62}$$

where $\overline{A}_{g',x}$ is gross carbon assimilation per unit leaf area for sunlit and shade leaves without photosynthesis down-regulation, $f_{dwn}$ is the factor for photosynthesis down-regulation, $\gamma_{gd}$ and $\gamma_g$ are parameters that characterize the response to increased $CO_2$, and $C_0$ is the base concentration of $CO_2$. The Eqs. 61 and 62 are based on Arora et al. (2009), although the original MATSIRO does not consider the effect of photosynthesis down-regulation. We set $\gamma_{gd} = 0.42$, $\gamma_g = 0.9$, and $C_0 = 288$ according to





Arora et al. (2009). The calculation for $\overline{A}_{g',x}$ and $\overline{R}_{d,x}$ is based on the leaf photosynthesis model developed by Collatz et al. (1991). In their model, $\overline{A}_{g',x}$ is determined by three limiting factors: Rubisco, light, and sucrose synthesis, as follows:

$$\overline{A}_{g',x} \leq \min\left(\overline{\omega}_{c,x}, \overline{\omega}_{e,x}, \overline{\omega}_{s,x}\right), \tag{63}$$

where $\overline{\omega}_{c,x}$, $\overline{\omega}_{e,x}$, and $\overline{\omega}_{s,x}$ are Rubisco-limited, light-limited, and sucrose-limited carbon assimilation per unit leaf area, respectively. To implement smooth transition between each limited state, $\overline{A}_{g',x}$ is determined practically by solving the following two equations (Sellers et al., 1996b):

$$\beta_{ce}\overline{\omega}_{p,x}^2 - \overline{\omega}_{p,x}^2(\overline{\omega}_{c,x}^2 + \overline{\omega}_{e,x}^2) + \overline{\omega}_{c,x}^2\overline{\omega}_{e,x}^2 = 0 \tag{64}$$

$$\beta_{ps}\overline{A}_{g',x}^2 - \overline{A}_{g',x}^2(\overline{\omega}_{p,x}^2 + \overline{\omega}_{s,x}^2) + \overline{\omega}_{p,x}^2\overline{\omega}_{s,x}^2 = 0, \tag{65}$$

where $\beta_{ce}$ and $\beta_{pc}$ are the parameters that determine the smoothness of transition between each limited state. $\beta_{ce}$ is a crop-specific parameter and $\beta_{pc}$ is a parameter that does not depend on crop type. The variables $\overline{\omega}_{c,x}$, $\overline{\omega}_{e,x}$, and $\overline{\omega}_{s,x}$ are given by

$$\overline{\omega}_{c,x} = \overline{V}_{mc,x}\left\{\frac{c_{i,x} - \Gamma^*}{c_{i,x} + K_c(1 + [O_2]/K_O)}\right\} \tag{66}$$

$$\overline{\omega}_{e,x} = \epsilon_e\overline{Q}_x\left\{\frac{c_{i,x} + \Gamma^*}{c_{i,x} + 2\Gamma^*}\right\} \tag{67}$$

$$\overline{\omega}_{s,x} = \overline{V}_{ms,x}/2. \tag{68}$$

Here, $\overline{V}_{mc,x}$ and $\overline{V}_{ms,x}$ are the maximum Rubisco capacity per unit leaf area for $\overline{\omega}_{c,x}$ and $\overline{\omega}_{s,x}$, respectively, $c_{i,x}$ is the partial pressure of intercellular $CO_2$, $[O_2]$ is the partial pressure of intercellular $O_2$, $\overline{Q}_x$ is the photon flux density for PAR absorbed per unit leaf area by sunlit and shade leaves, $\epsilon_e$ is the quantum efficiency, $\Gamma^*$ is the light compensation point, and $K_c$ and $K_O$ are the Michaelis constant for $CO_2$ fixation and oxygen inhibition, respectively. We set $[O_2] = 20,900$ (Collatz et al., 1991). $\epsilon_e$ is a crop specific parameter. $\overline{V}_{mc,x}$ and $\overline{V}_{ms,x}$ are given by

$$\overline{V}_{mc,x} = \overline{V}_{max,x}[2^{Q_t}/\{1 + \exp\left(s_1(T_c - s_2)\right)\}], \tag{69}$$

$$\overline{V}_{ms,x} = \overline{V}_{max,x}[2^{Q_t}/\{1 + \exp\left(s_3(s_4 - T_c)\right)\}], \tag{70}$$

where $\overline{V}_{max,x}$ is the reference value for the maximum Rubisco capacity per unit leaf area of sunlit ($\overline{V}_{max,sn}$) and shade ($\overline{V}_{max,sh}$) leaves, $s_1$, $s_2$, $s_3$, and $s_4$ are parameters that represent temperature dependence of $\overline{V}_{max,x}$ on $\overline{V}_{mc,x}$ or $\overline{V}_{ms,x}$. The variables $s_1$ and $s_2$ are parameterised in Masutomi et al. (2016), whereas $s_3$ is a parameter that does not depend on crop type and $s_4$ is a crop-specific parameter. $Q_t$ is given by

$$Q_t = (T_c - 298)/10. \tag{71}$$




$\overline{V}_{max,sn}$ and $\overline{V}_{max,sh}$ are defined by

$$\overline{V}_{max,sn} = \left( \int_0^L V_{max}(l) f_{sn}(l) dl \right) / L_{sn}, \tag{72}$$

$$\overline{V}_{max,sh} = \left( \int_0^L V_{max}(l)(1 - f_{sn}(l)) dl \right) / L_{sh}, \tag{73}$$

where $V_{max}(l)$ is the reference value for the maximum Rubisco capacity at a LAI depth of $l$. The vertical distribution of $V_{max}(l)$ depends on that of leaf nitrogen within canopy and is given by

$$V_{max}(l) = V_{max}(0) \exp(-K_n l), \tag{74}$$

where $K_n$ is a parameter that represents the vertical distribution of leaf nitrogen, and $V_{max}(0)$ is the reference value for the maximum Rubisco capacity at the canopy top. $V_{max}(0)$ as well as $s_1$ and $s_2$ are parameterized in Masutomi et al. (2016), and we set $K_n = 0.3$ (Oleson and Lawrence, 2013). $\Gamma^*$, $K_c$, and $K_O$ are given by

$$\Gamma^* = 0.5[O_2]/S, \tag{75}$$

$$K_c = 30 \times 2.1^{Q_t}, \tag{76}$$

$$K_O = 30000 \times 1.2^{Q_t}, \tag{77}$$

$$S = 2600 \times 0.57^{Q_t}, \tag{78}$$

where $S$ is the ratio of the partition of RuBP to the caboxylase or oxygenase reactions of Rubisco.

$\overline{Q}_x$ in Eq. (67) is defined by the following equation:

$$\overline{Q}_x = Q_x / L_x. \tag{79}$$

Here, $Q_x$ is the PAR absorbed by the entire canopy for sunlit ($Q_{sn}$) and shade ($Q_{sh}$) leaves. $Q_{sn}$ and $Q_{sh}$ consist of direct and scattered components and are given as

$$Q_{sn} = Q_{sn,d} + Q_{sn,s}, \tag{80}$$

$$Q_{sh} = Q_{sh,s}, \tag{81}$$

where $Q_{sn,d}$, $Q_{sn,s}$, and $Q_{sh,s}$ are the direct PAR absorbed by sunlit leaves, the scattered PAR absorbed by sunlit leaves, and the scattered PAR absorbed by shade leaves, respectively. These are described by

$$Q_{sn,d} = k_q \int_0^L \frac{dD_1^d(l)}{dl} dl, \tag{82}$$

$$Q_{sn,s} = k_q \int_0^L \frac{d(S_1^d(l) - S_1^u(l))}{dl} f_{sn}(l) dl, \tag{83}$$

$$Q_{sh,s} = k_q \int_0^L \frac{d(S_1^d(l) - S_1^u(l))}{dl} (1 - f_{sn}(l)) dl, \tag{84}$$





where $D_1^d(l)$, $S_1^d(l)$, and $S_1^u(l)$ are calculated by the LSM (Eqs. 12 to 14) and $k_q$ is a constant that transfers the radiant flux density to photon flux density.

$\overline{R}_{d,x}$ in Eq. 57 is given by the following equation:

$$\overline{R}_{d,x} = f_d \overline{V}_{max,x}[2^{Q_t}/\{1 + \exp\left(s_5(T_c - s_6)\right)\}], \tag{85}$$

where $f_d$ is a respiration factor and crop-specific parameter, whereas $s_5$ and $s_6$ are parameters that are not crop-dependent. It should be noted that $\overline{A}_{n,x}$ can be calculated using the equations described in this section (Eqs. 57 to 85) if $c_{i,x}$ is given.

$\overline{A}_{n,x}$ should be equal to the $CO_2$ flux between the leaf interior and boundary layer and the $CO_2$ flux between the leaf boundary layer and the atmosphere. If these requirements are fulfilled the following equation can be derived:

$$\overline{A}_{n,x} = (\overline{g}_l/P_a)(c_a - c_{s,x})/1.4 = (\overline{g}_{st,x}/P_a)(c_{s,x} - c_{i,x})/1.6, \tag{86}$$

where $c_a$ is the partial pressure of atmospheric $CO_2$, $c_{s,x}$ is the partial pressure of $CO_2$ at the leaf boundary layer for sunlit and shade leaves, $\overline{g}_l$ is the leaf boundary conductance for vapour per unit leaf area, and $\overline{g}_{st,x}$ is the stomatal conductance for vapour per unit leaf area for sunlit and shade leaves. From Eq. 86, $c_{i,x}$ and $c_{s,x}$ are defined by

$$c_{i,x} = c_a - (1.4/\overline{g}_l + 1.6/\overline{g}_{st,x})\overline{A}_{n,x}P_a, \tag{87}$$

$$c_{s,x} = c_a - 1.4\overline{A}_{n,x}P_a/\overline{g}_l. \tag{88}$$

The parameters $c_a$ and $\overline{g}_l$ are given by

$$c_a = (C_a * 10^{-6})P_a, \tag{89}$$

$$\overline{g}_l = (\overline{g}_a/2) * P_a/(T_c R_{vap}\omega_{H_2O}), \tag{90}$$

$$\overline{g}_a = c_h U_c. \tag{91}$$

where $w_{H_2O}$ is a constant for the molar weight of vapour, $\overline{g}_a$ is the leaf boundary conductance for heat per unit leaf area (for both sides of the leaf), $c_h$ is the leaf transfer coefficient for heat and is a crop specific parameter, $U_c$ is the mean wind speed in the canopy (Appendix F). Note that Eqs. 90 and 91 are based on Maruyama and Kuwagata (2008), whereas the original MATSIRO uses $C_h$ instead of $\overline{g}_a/2$ in Eq. 90.

$\overline{A}_{n,x}$ meets the Ball-Berry relationship (Ball, 1988), which describes the relationship between $\overline{A}_{n,x}$, $\overline{g}_{st,x}$, and other environmental conditions. The Ball-Berry relationship is given by

$$\overline{g}_{st,x} = \begin{cases} m\dfrac{\overline{A}_{n,x}P_a}{c_{s,x}}h_{s,x} + b & (\text{if } \overline{A}_{n,x} > 0), \\ b & (\text{otherwise}) \end{cases} \tag{92}$$

where $m$ and $b$ are the slope and intercept of the Ball-Berry relationship, and $h_{s,x}$ is the relative humidity at leaf boundary. It is noteworthy that $b$ indicates the stomatal conductance when $\overline{A}_{n,x}$ is equal to or less than zero (Baldocchi, 1994) and that the effect of water stress on $b$ is not considered in MATCRO-Rice because the surface is flooded. The variables $m$ and $b$ are crop specific parameters, and $h_{s,x}$ is defined by

$$h_{s,x} = e_{s,x}/e_{sat}(T_c, P_a), \tag{93}$$





where $e_{s,x}$ is the vapour pressure at leaf boundary and $e_{sat}$ is the saturated vapour pressure. The variable $e_{s,x}$ is expressed as

$$e_{s,x} = (e_a \overline{g}_l + e_i \overline{g}_{st,x})/(\overline{g}_l + \overline{g}_{st,x}), \tag{94}$$

where $e_a$ and $e_i$ are the vapour pressure in the air and leaf, respectively. Eq. 94 is derived from the fact that the water vapour flux from the stomata to leaf surface is equal to the water vapour flux from the leaf surface into the atmosphere, which is shown in the following equation:

$$\overline{g}_{st,x}(e_i - e_s) = \overline{g}_l(e_{s,x} - e_a). \tag{95}$$

The parameters $e_a$, $e_i$, and $e_{sat}$ are given by

$$e_a = Q(R_{dry}/R_{vap}), \tag{96}$$

$$e_i = e_{sat}(T_c, P_a), \tag{97}$$

$$e_{sat}(T_c, P_a) = Q_{sat}(T_c, P_a)(R_{dry}/R_{vap}), \tag{98}$$

where $e_i$ is assumed to be saturated.

Now we have three relationships (Eqs. 57 to 85, Eq. 87, and Eq. 92) in terms of three unknown variables ($\overline{A}_{n,x}$, $c_{i,x}$, and $\overline{g}_{st,x}$). Therefore, we can determine the values for $\overline{A}_{n,x}$, $c_{i,x}$, and $\overline{g}_{st,x}$, by numerically solving the three relationships. The numerical method is described in Masutomi et al. (2016).

Last, $\overline{g}_s$ is given by the following equation:

$$\overline{g}_s = \overline{g}_{st} * (T_c R_{vap} w_{H_2O}/P_a), \tag{99}$$

$$\overline{g}_{st} = \{(\overline{g}_{st,sn} * L_{sn} + \overline{g}_{st,sh} * L_{sh})/L\} * 2, \tag{100}$$

where $\overline{g}_{st}$ is the stomatal conductance for vapour per unit leaf area for both sides of the leaf.

## 4.2 Crop development

The crop development module calculates $DVS$, which is an index used to quantify developmental stage of crops. $DVS$ is mainly used for determining the timing of transplanting, heading, and harvesting. In addition, $DVS$ is used for partitioning of carbon assimilation into each organ and for estimating LAI and height. This module is based on the formulation by Bouman et al. (2001). $DVS$ is calculated from

$$DVS = GDS/mGDS, \tag{101}$$

$$GDS = \int_0^t DVR\, dt', \tag{102}$$

$$DVR = \begin{cases} 0 & (T_a < T_b | T_h \leq T_a) \\ T_a - T_0 & (T_b \leq T_a < T_o) \\ (T_o - T_b)(T_h - T_a)/(T_h - T_o) & (T_o \leq T_a < T_h) \end{cases}, \tag{103}$$





where $GDS$ is the growing degree seconds at $t$, $mGDS$ is $GDS$ required until maturation, $DVR$ is the development rate at $t$, $T_0$ is the melting temperature of water, and $T_b$, $T_h$, and $T_o$ are the minimum temperature, maximum temperature, and optimal temperature for development, respectively. The value of $mGDS$ is parameterized in Masutomi et al. (2016), and $T_b$, $T_h$, and $T_o$ are crop-specific parameters. $T_0$ is a physical constant (Table 4). It should be noted that $DVS = 0$ represents sowing and $DVS = 1$ represents maturation. Furthermore, we introduce two parameters that represent the timing of emergence ($eDVS$) and heading ($hDVS$). Both $eDVS$ and $hDVS$ are crop-specific parameters. The values of $eDVS$ and $hDVS$ are parameterized in Masutomi et al. (2016).

During the transplantation of rice seedling, the seedlings enter transplanting shock, which prevents shoot growth (Bouman et al., 2001). In MATCRO-Rice, the transplanting shock period is defined by $DVS$, where $trDVS$ is DVS at the time when transplanting shock starts and $teDVS$ is DVS at which transplanting shock ends. Both $trDVS$ and $teDVS$ are parameterized in Masutomi et al. (2016).

### 4.3 Crop growth

This module calculates the growth of organs and reserves. The organs considered in MATCRO-Rice include leaf, stem, panicle, and root. In addition, the model considers glucose reserves in leaves and starch reserves in stem. All carbon assimilated in leaves through photosynthesis is first stored in leaf in the form of glucose. Then, the stored glucose is partitioned to each organ and stored in the stem when the amount of the stored glucose exceeds the critical rate to dry weight of leaf. This module is based on MACROS (Penning de Vries et al., 1989).

The dry weights of each organ and reserve are expressed by

$$W_{lef} = W_{lef,0} + \int_{t_e}^{t} (G_{R,lef} - L_{S,lef}) dt', \tag{104}$$

$$W_{stm} = W_{stm,0} + \int_{t_e}^{t} G_{R,stm} dt', \tag{105}$$

$$W_{pnc} = \int_{t_e}^{t} G_{R,pnc} dt' \tag{106}$$

$$W_{rot} = W_{rot,0} + \int_{t_e}^{t} G_{R,rot} dt', \tag{107}$$

$$W_{stc} = \int_{t_e}^{t} (G_{R,stc} - R_{M,stc}) dt', \tag{108}$$

$$W_{glu} = W_{glu,0} + \int_{t_e}^{t} G_{R,glu} dt', \tag{109}$$



where $W_{lef}$, $W_{stm}$, $W_{pnc}$, $W_{rot}$, $W_{stc}$, $W_{glu}$ are the dry weight of leaves, stems, panicles, roots, starch reserves, and glucose reserves at $t$, respectively, $W_{lef,0}$, $W_{stm,0}$, $W_{rot,0}$, and $W_{glu,0}$ represent the initial dry weight at emergence of each organ and reserve, $G_{R,lef}$, $G_{R,stm}$, $G_{R,pnc}$, $G_{R,rot}$, $G_{R,stc}$, and $G_{R,glu}$ are the growth rates of the corresponding organ and reserve, $L_{S,lef}$ is the loss rate of leaves due to leaf death, $R_{M,stc}$ is the loss rate of starch reserves in stem due to remobilization, $t_e$ is

the time at emergence after sowing, and $W_{lef,0}$, $W_{stm,0}$, $W_{rot,0}$, and $W_{glu,0}$ are simulation setting parameters.

The glucose reserve in leaf is supplied through photosynthesis in leaves and remobilization from the stem. Thus, the supply of glucose is given by

$$S_{glu} = A_n C_{CO2,glu} + R_{M,stc} C_{stc,glu}, \tag{110}$$

where, $S_{glu}$ is the supply of glucose to leaf reserve, $A_n$ is the net carbon assimilation calculated in Eq. 56, and $C_{CO_2,glu}$ and

$C_{stc,glu}$ are the conversion factors from $CO_2$ or starch to glucose, which are chemically determined (Table 4). We assumed that the partition of glucose in leaves to each organ occurs if the following equation is met:

$$W_{glu} + S_{glu}\delta t > k_{glu}W_{lef}, \tag{111}$$

where $\delta t$ is one simulation time step, $k_{glu}$ is the critical ratio at which the partition of glucose happens, and $\delta t$ is a simulation setting parameter. We set $k_{glu} = 0.1$ (Penning de Vries et al., 1989). When Eq. 111 is met, the amount of glucose that exceeds

the critical ratio is partitioned to each organ and reserve according to the following equation:

$$G_{P,glu} = (W_{glu} + S_{glu}\delta t - k_{glu}W_{lef})/\delta t, \tag{112}$$

where $G_{P,glu}$ is the amount of glucose partitioned to each organ and reserve. The growth rate of each organ and reserve is expressed as follows:

$$
\begin{aligned}
G_{R,lef} &= G_{P,glu}P_{R,sh}P_{R,lef}C_{glu,lef}, & (113)\\
G_{R,stm} &= G_{P,glu}P_{R,sh}(1 - P_{R,lef} - P_{R,pnc})(1 - f_{stc})C_{glu,stm}, & (114)\\
G_{R,pnc} &= G_{P,glu}P_{R,sh}P_{R,pnc}C_{glu,pnc}, & (115)\\
G_{R,rot} &= G_{P,glu}(1 - P_{R,sh})C_{glu,rot}, & (116)\\
G_{R,stc} &= G_{P,glu}P_{R,sh}(1 - P_{R,lef} - P_{R,pnc})f_{stc}C_{glu,stc}, & (117)\\
G_{R,glu} &= (k_{glu}W_{lef} - W_{glu})/\delta t, & (118)
\end{aligned}
$$

where $P_{R,sh}$ is the ratio of glucose partitioned to shoot, $P_{R,lef}$ and $P_{R,pnc}$ are the partition ratios of glucose from shoot to leaf and panicle, $f_{stc}$ is the proportion of glucose allocated to starch reserve in stem, $C_{glu,lef}$, $C_{glu,stm}$, $C_{glu,rot}$, $C_{glu,pnc}$, and $C_{glu,stc}$ are dry weight of corresponding organs and reserves that are produced from the unit weight of glucose. $f_{stc}$, $C_{glu,lef}$, $C_{glu,stm}$, $C_{glu,rot}$, and $C_{glu,pnc}$ are crop-specific parameters. $f_{stc}$ is parameterized in Masutomi et al. (2016). We set the values of $C_{glu,lef}$, $C_{glu,stm}$, $C_{glu,rot}$, and $C_{glu,pnc}$ according to Penning de Vries et al. (1989). $C_{glu,stc}$ is a chemical

constant. If Eq. 111 is not met, glucose is not partitioned into each organ and reserve, except as the glucose reserve in leaf.





Therefore, the growth rate of each organ and reserve are calculated as follows:

$$G_{R,lef} = G_{R,stm} = G_{R,rot} = G_{R,pnc} = G_{R,stc} = 0 \tag{119}$$

$$G_{R,glu} = S_{glu}. \tag{120}$$

The partition ratios to each organ are given as

$$
P_{R,sh} = \begin{cases}
1 - P_{rot} & (DVS \leq trDVS) \\
0 & (trDVS < DVS \leq teDVS) \\
1 - P_{rot} & (teDVS < DVS \leq DVS_{rot1}) \\
\frac{1 - P_{rot}(DVS_{rot1} - DVS)}{(DVS_{rot2} - DVS_{rot1})} & (DVS_{rot1} < DVS \leq DVS_{rot2}) \\
1 & (\text{Otherwise})
\end{cases}, \tag{121}
$$

$$
P_{R,lef} = \begin{cases}
P_{lef} & (DVS \leq DVS_{lef1}) \\
\frac{P_{lef}(DVS_{lef2} - DVS)}{(DVS_{lef2} - DVS_{lef1})} & (DVS_{lef1} < DVS \leq DVS_{lef2}), \\
0 & (\text{Otherwise})
\end{cases} \tag{122}
$$

$$
P_{R,pnc} = \begin{cases}
0 & (DVS \leq DVS_{pnc1}) \\
\frac{(DVS - DVS_{pnc1})}{(DVS_{pnc2} - DVS_{pnc2})} & (DVS_{pnc1} < DVS \leq DVS_{pnc2}), \\
1 & (\text{Otherwise})
\end{cases} \tag{123}
$$

where $DVS_{rot1}$, $DVS_{rot2}$, $DVS_{lef1}$, $DVS_{lef2}$, $DVS_{pnc1}$, and $DVS_{pnc2}$ represent the $DVS$ values at which corresponding partitions change, $P_{rot}$ is the ratio of partitioned glucose to the roots at $DVS < DVS_{rot1}$, and $P_{lef}$ is the ratio of glucose partitioned to the leaf and glucose partitioned to shoot at $DVS < DVS_{lef}$. $DVS_{rot1}$, $DVS_{rot2}$, $DVS_{lef1}$, $DVS_{lef2}$, $DVS_{pnc1}$, $DVS_{pnc2}$, $P_{rot}$, and $P_{lef}$ are crop-specific parameters and are parameterized in Masutomi et al. (2016). In Eq. 121, we assume that no glucose is partitioned to shoot during transplanting shock ($teDVS < DVS \leq teDVS$). It is important to note that transplanting shock is considered only when transplanting is conducted.

Loss of leaf dry weight due to leaf death ($L_{S,lef}$) and remobilization from starch reserve in stem ($R_{M,stm}$) occur after heading and they are defined as follows

$$
L_{S,lef} = \begin{cases}
0 & (DVS \leq hDVS), \\
r_{dd,lef}(W_{lef} + W_{glu}) & (\text{Otherwise})
\end{cases} \tag{124}
$$

$$
R_{M,stc} = \begin{cases}
0 & (DVS \leq hDVS), \\
r_{rm,stc}W_{stc} & (\text{Otherwise})
\end{cases} \tag{125}
$$





where $r_{dd,lef}$ and $r_{rm,stc}$ represent the ratios of leaf death and remobilization. $r_{dd,lef}$ varies with $DVS$ as follow:

$$r_{dd,lef} = r_{d1,lef}(DVS - hDVS)/(1 - hDVS) \tag{126}$$

where $r_{d1,lef}$ is the ratio of leaf death at harvest ($DVS = 1$) and it is parameterized in Masutomi et al. (2016). We set $r_{rm,stc} = 1.16 * 10^{-6}$, assuming that all starch stored in stem is remobilized in 10 days after heading (Bouman et al., 2001).

Last, the dry weight of shoot ($W_{sh}$), used in Section 3.4, is given by

$$W_{sh} \quad = \quad W_{lef} + W_{stm} + W_{pnc} + W_{stc} + W_{glu}. \tag{127}$$

## 4.4   LAI, crop height, and root depth

Leaf area index ($L$), crop height ($h_{gt}$), and root depth ($z_{rt}$) are expressed as

$$L \quad = \quad (W_{lef} + W_{glu})/SLW, \tag{128}$$

$$SLW \quad = \quad SLW_{mx} + (SLW_{mn} - SLW_{mx})\exp(-k_{SLW}DVS), \tag{129}$$

$$h_{gt} \quad = \quad \begin{cases} h_{gt,aa}L^{h_{gt,ab}} & (DVS < hDVS), \\ h_{gt,ba}L^{h_{gt,bb}} & (hDVS < DVS) \end{cases} \tag{130}$$

$$z_{rt} \quad = \quad \min\{z_{rt,mx}, r_{rt}(t - t_e)\}, \tag{131}$$

where $SLW$ is the specific leaf weight, $SLW_{mx}$ and $SLW_{mn}$ are the maximum and minimum values of specific leaf weight, respectively, $k_{SLW}$ is a parameter that determines the relationship between $DVS$ and specific leaf weight, $h_{gt,aa}, h_{gt,ab}, h_{gt,ba}$, and $h_{gt,bb}$ are parameters that define the relationship between LAI and crop height, $z_{rt,mx}$ is the maximum root depth, and $r_{rt}$ is the root growth rate. The allometric equations for estimating crop height (Eq. 130) is based on Maruyama and Kuwagata (2010). $SLW_{mx}, SLW_{mn}, k_{SLW}, h_{gt,aa}, h_{gt,ab}, h_{gt,ba}$,and $h_{gt,bb}$ are crop-specific parameters; they are parameterized in Masutomi

et al. (2016). $z_{rt,mx}$ and $r_{rt}$ are also crop-specific parameters, and they are set to $z_{rt,mx} = 0.3$ and $r_{rt} = 1.16 * 10^{-7} (= 0.01$ m day$^{-1}$) (Penning de Vries et al., 1989).

## 4.5   Crop yield

Crop yield is calculated from dry weight of the panicle at maturity as follows:

$$Yld = k_{yld}W_{pnc,mt}, \tag{132}$$

where $Yld$ is the crop yield, $W_{pnc,mt}$ is the dry weight of the panicle at maturity, $k_{yld}$ is the ratio of the crop yield to $W_{pnc,mt}$. The variable $k_{yld}$ is a crop specific parameter and it is parameterized in Masutomi et al. (2016).





## 5   Concluding remarks

We developed a new LSM-CGM combined model for paddy rice fields called MATCRO-Rice, which is fully described in the present paper. MATCRO-Rice has two features: (i) The model can consistently simulate LHF, SHF, biomass growth for each organ, and crop yield by exchanging variables listed in Table 2; (ii) The model considers water surface in paddy rice fields.

According to our literature survey, MATCRO-Rice is the first LSM-CGM combined model for rice that employs these two features.

The first feature enables us to apply the model to a wide range of integrated issues. For example, by using MATCRO-Rice, we can assess the impacts of paddy rice fields on climate through heat and water fluxes and consistently assess the impacts of climate on rice productivity. Osborne et al. (2009) showed that the interaction between agricultural land and climate can

play an important role in the annual variability of both the climate and crop yield. MATCRO-Rice can investigate the impact of the interactions at paddy rice fields on climate and rice productivity. MATCRO-Rice can be a useful tool for addressing the integrated issues of agriculture and hydrology.

MATCRO-Rice can be also applied to simultaneously assess the climate change impacts on rice productivity and hydrological cycle in paddy rice fields. Masutomi et al. (2009) showed that climate change will have significant impact on rice

productivity across Asia. In addition, agricultural land is one of the key players in global hydrological cycle, and climate change will alter globally the hydrological cycle (Oki and Kanae, 2006).

The first feature also gives us a chance to comprehensively evaluate the model with observations (Lei et al., 2010). Model evaluation is described in the companion paper (Masutomi et al., 2016).

The current version (Ver. 1) of MATCRO-Rice has a couple of major limitations. First, nitrogen dynamics is not included in

MATCRO-Rice, although it is well known that nitrogen stress significantly affects crop growth, and hence LHF and SHF. This indicates that MATCRO-Rice simulates LHF, SHF, biomass growth, and crop yield with no nitrogen stress. To apply the model to the site with nitrogen stress, it is necessary to include nitrogen dynamics. This feature is an important future challenge.

Second, the impact of water stress on crop growth is not considered in MATCRO. This limitation is not considered a problem in irrigated land but in rain-fed land. If the model is applied in rain-fed lands, the model needs to be improved.

## 25   6   Code availability

The source code of MATCRO will be distributed at request to the corresponding author (Yuji Masutomi: yuji.masutomi@gmail.com). The website for MATCRO-Rice will be developed in the near future.





## Appendix A: $\rho_a$ and $Q_{sat}$

The air density ($\rho_a$) and the specific humidity at saturation ($Q_{sat}$) are calculated physically according to the equation for the state of dry air and the Clausisu-Clapeyron equation, respectively, as follow:

$$\rho_a = P_a/(R_{dry}T_a), \tag{A1}$$

$$Q_{sat}(T_x, P_a) = (R_{dry}/R_{vap})\{e_{sat}(T_0)\exp\left((\lambda/R_{vap})(1/T_0 - 1/T_x)\right)\}/P_a, \tag{A2}$$

where $T_a$ is air temperature, $P_a$ is air pressure, $T_x$ is temperature of the canopy ($T_c$) or surface water ($T_w$), $T_0$ is the melting temperature of the water, $R_{dry}$ and $R_{vap}$ are the gas constants of the dry air and vapour, respectively, $e_{sat}(T_0)$ is the vapour pressure at melting temperature of the water, and $\lambda$ is the latent heat of vaporisation. $T_a$ and $P_a$ are meteorological inputs (Table 1). $T_x$ ($T_c$ or $T_w$) is calculated in Section 3.1. The other parameters are physical constants (Table 4).

## Appendix B: Zenith angle $\theta$

According to Goudriaan and van Laar (1994), zenith angle of the sun ($\theta$) is calculated as follows.

$$\cos(\theta) = \sin(2\pi L_t/360)\sin(\delta_s) + \cos(2\pi L_t/360)\cos(\delta_s)\cos(h_{arg}), \tag{B1}$$

$$\delta_s = -\arcsin(\sin(23.45(2\pi/360))\cos(2\pi(D_{oy}+10)/365)), \tag{B2}$$

$$h_{arg} = 2\pi(h_r - 12)/24, \tag{B3}$$

where $L_t$ is the latitude in radians at the simulation site, $\delta_s$ is the declination of the sun, $h_{arg}$ is the hour angle from noon ($h_r = 12$), $D_{oy}$ is the number of days from Jan 1 at the simulation site, and $h_r$ is the local time at the simulation site.





## Appendix C: Coefficients for radiation equations

The coefficients for radiation equations (Eqs. 12–14) are calculated as follows:

$$a_i = Fd_f\{(1-t_i)^2 - r_i^2\}^{1/2}, \tag{C1}$$

$$
\begin{aligned}
C_{1,i} = \{&-(A_{2,i} - r_g)(S_i^d(0) - C_{3,i}D_i^d(0))\exp(-a_iL) \\
&+(C_{3,i}r_g + r_g - C_{4,i})D_i^d(0)\exp(-FL\sec(\theta)))\}/A_{3,i},
\end{aligned}
\tag{C2}
$$

$$
\begin{aligned}
C_{2,i} = \{&(A_{1,i} - r_g)(S_i^d(0) - C_{3,i}D_i^d(0))\exp(a_iL), \\
&-(C_{3,i}r_g + r_g - C_{4,i})D_i^d(0)\exp(-FL\sec(\theta)))\}/A_{3,i},
\end{aligned}
\tag{C3}
$$

$$C_{3,i} = \sec(\theta)\{t_i\sec(\theta) + d_f t_i(1-t_i) + d_f r_i^2\}/\{d_f^2((1-t_i)^2 - r_i^2) - \sec^2(\theta)\}, \tag{C4}$$

$$C_{4,i} = \{r_i(d_f - \sec(\theta))\sec(\theta))\}/\{d_f^2((1-t_i)^2 - r_i^2) - \sec^2(\theta)\}, \tag{C5}$$

$$A_{1,i} = (1 - t_i + \{(1-t_i)^2 - r_i^2\}^{1/2})/r_i, \tag{C6}$$

$$A_{2,i} = (1 - t_i - \{(1-t_i)^2 - r_i^2\}^{1/2})/r_i, \tag{C7}$$

$$A_{3,i} = (A_{1,i} - r_g)\exp(a_iL) - (A_{2,i} - r_g)\exp(-a_iL), \tag{C8}$$

where $i$ indicates the wavebands of radiation ($i = 1$: PAR; $i = 2$: NIR), $r_i$ and $t_i$ are the leaf reflectivity and transmissivity, respectively, $F$ is the distribution of leaf orientation, $d_f$ is a scattering factor, $A_{3,i}$ is a new variable introduced in Eqs. C2 and C3, $L$ is the LAI, $r_g$ is the surface albedo for shortwave radiation, $D_i^d(0)$ and $S_i^d(0)$ are direct and scattered downward radiant flux density at the canopy top, respectively, and $\theta$ is the zenith angle of the sun. $r_i$ and $t_i$ are crop-specific parameters determined by Sellers et al. (1996b). $F$ is set to 0.5 from the assumption of random leaf orientation (Goudriaan and van Laar, 1994), and $d_f$ is $\sec(2\pi(53/360))$ (Watanabe and Ohtani, 1995). $A_{3,i}$ is defined in Eq. C8, $L$ is calculated in the CGM (Eq. 128), and $r_g$ for surface water is given in Maruyama and Kuwagata (2010). $D_i^d(0)$ and $S_i^d(0)$ are given in Eqs. 15 and 16, respectively, and $\theta$ is calculated in B1.

It should be noted that $a_i$, $A_{1,i}$, and $A_{2,i}$ are not variables determined by constant parameters, while $C_{1,i}$, $C_{2,i}$, $C_{3,i}$, $C_{4,i}$, and $A_{3,i}$ are variables.

## Appendix D: Reflectivity and transmissivity of canopies

Reflectivity ($r_{ij}$) and transmissivity ($\tau_{ij}$) of canopy for each waveband ($i = 1$: PAR, $i = 2$: NIR) and for each direction ($j = 1$: direct, $j = 2$: scattered) are given as follows.

$$r_{i1} = C_{4,i} - C_{3,i}r_{i2}, \tag{D1}$$

$$r_{i2} = (A_{1,i}C_{1,i} + A_{2,i}C_{2,i})/(C_{1,i} + C_{2,i}), \tag{D2}$$

$$\tau_{i1} = (1 + C_{3,i} - C_{4,i}\exp(-FL\sec(\theta))) - C_{3,i}\tau_{i2}, \tag{D3}$$

$$\tau_{i2} = \{(C_{1,i}(1 - A_{1,i})\exp(a_iL)) + C_{2,i}(1 - A_{2,i}\exp(-a_iL))\}/(C_{1,i} + C_{2,i}), \tag{D4}$$





where $a_i$, $C_{1,i}$, $C_{2,i}$, $C_{3,i}$, $C_{4,i}$, $A_{1,i}$, and $A_{2,i}$, the coefficients of radiation equations (Eqs. 12–14), are calculated as shown in Appendix C, $F$ is a parameter that defines the distribution of leaf orientation, $L$ is the LAI, which is calculated in the CGM (Eq. 128), and $\theta$ is the zenith angle of the sun (Appendix B).

**Appendix E:  $d, z_M, z_T, z_Q, z_{Mw}, z_{Tw},$ and $z_{Qw}$**

Zero-plane displacement height ($d$), roughness lengths of an entire surface for the profiles of momentum, temperature, and specific humidity ($z_M$, $z_T$, and $z_Q$), and roughness lengths that express the effect of surface water on the profiles of momentum, temperature, and specific humidity ($z_{Mw}$, $z_{Tw}$, and $z_{Qw}$) are calculated according to Watanabe (1994) as follows.

$$d = h_{gt}\left[1 - \frac{1}{A^+}\{1 - \exp(-A^+)\}\right], \tag{E1}$$

$$\left(\ln\frac{h_{gt} - d}{z_M}\right)^{-1} = \left\{1 - \exp(-A^+) + \left(-\ln\frac{z_{Ms}}{h_{gt}}\right)^{-1/0.45}\exp(-2A^+)\right\}^{0.45}, \tag{E2}$$

$$\left(\ln\frac{h_{gt} - d}{z_M}\right)^{-1}\left(\ln\frac{h_{gt} - d}{z_X}\right)^{-1} = C_X^\infty\left\{1 - \exp(-P_{3X}A^+) + \left(\frac{C_X^0}{C_X^\infty}\right)^{1/0.9}\exp(-P_{4X}A^+)\right\}^{0.9}, \tag{E3}$$

$$\left(\ln\frac{h_{gt} - d}{z_{Mw}}\right)^2 = \left(\ln\frac{h_{gt} - d}{z_M}\right)\left(\ln\frac{h_{gt} - d}{z_M^+}\right), \tag{E4}$$

$$\left(\ln\frac{h_{gt} - d}{z_{Mw}}\right)\left(\ln\frac{h_{gt} - d}{z_{Xw}}\right) = \left(\ln\frac{h_{gt} - d}{z_M}\right)\left(\ln\frac{h_{gt} - d}{z_X^+}\right), \tag{E5}$$

$$A^+ = \frac{c_m L}{2\kappa^2}, \tag{E6}$$

$$C_X^0 = \left(\ln\frac{h_{gt} - d}{z_M}\right)^{-1}\left(\ln\frac{h_{gt} - d}{z_X^+}\right)^{-1}, \tag{E7}$$

$$C_X^\infty = \frac{-1 + (1 + 8F_X)^{0.5}}{2}, \tag{E8}$$

$$F_X = \frac{c_X}{c_m}, \tag{E9}$$

$$\left(\ln\frac{h_{gt} - d}{z_*^+}\right)^{-1} = \frac{1}{-\ln\left(\frac{z_{*s}}{h_{gt}}\right)}\left(\frac{P_{1*}}{P_{1*} + A^+\exp(A^+)}\right)^{P_{2*}}, \tag{E10}$$

$$P_{1*} = 0.00115\left(\frac{z_{*s}}{h_{gt}}\right)^{0.1}\exp\left\{5\left(\frac{z_{*s}}{h_{gt}}\right)\right\}, \tag{E11}$$

$$P_{2*} = 0.55\exp\left\{-0.58\left(\frac{z_{*s}}{h_{gt}}\right)^{0.35}\right\}, \tag{E12}$$

$$P_{3X} = \{F_X + 0.084\exp(-15F_X)\}^{0.15}, \tag{E13}$$

$$P_{4X} = 2F_X^{1.1}, \tag{E14}$$

$$c_e = c_h/(1 + c_h(U_c/\bar{g}_s)). \tag{E15}$$




Here, $z_{Ms}, z_{Ts}$, and $z_{Qs}$ are the roughness lengths of surface water for momentum, temperature, and specific humidity, respectively. In this model, we assume $z_{Ms}, z_{Ts}$, and $z_{Qs} = 0.001$ m (Kimura and Kondo, 1998). $c_m$, $c_h$, and $c_e$ are the leaf transfer coefficients for momentum, temperature, and specific humidity, respectively. $c_m$ an $c_h$ are crop-specific parameters, while $c_e$ is calculated in Eq. E15. $h_{gt}$ and $L$ are crop height and LAI, respectively, and are calculated in the CGM (Eqs. 130 and 128).

$\overline{g}_s$ is the stomatal conductance per unit leaf area for both sides of the leaf (Eq. 99). $U_c$ is the mean wind speed in the canopy and is calculated in Appendix F. $A^+$, $C_X^0$, $C_X^\infty$, $z_M^+$, $z_X^+$, $z_*^+$, $P_{1*}$, $P_{2*}, P_{3X}$, $P_{4X}$, $F_X$ are the intermediate variables, and $\kappa$ is the Karman constant. The symbol "$*$" indicates "M", "T", or "Q", and the symbol "X" indicates "T" or "Q".

**Appendix F:   Mean wind speed in the canopy**

Mean wind speed in the canopy ($U_c$) is expressed as

$$U_c = (U_h/\gamma_m h_{gt}) * \{1 - \exp(-\gamma_m h_{gt})\}, \tag{F1}$$
$$U_h = U/(1 + \ln((z_a - h_{gt}) + 1)), \tag{F2}$$
$$\gamma_m = c_m(L/h_{gt})/(2k^2), \tag{F3}$$

where $U_h$ is the reference wind speed, and $\gamma_m$ is the coefficient of exponential decrease for wind speed in the canopy.

*Acknowledgements.* We would like to acknowledge Drs. Kuawagata, T. and Kim, W. at NIAES for useful discussion about land surface
modelling. We are also grateful to Mrs Hatanaka for her help in extensive literature survey. This research was supported by the Environment
Research and Technology Development Fund (S-12) and the Program on Development of Regional Climate Change Adaptation Plans in
Indonesia (PDRCAPI) of the Ministry of the Environment.



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



**Table 1.** Meteorological inputs

| Variable | Unit | Description |
| --- | --- | --- |
| $P_a$ | Pa | Air pressure |
| $P_r$ | kg m$^{-2}$ s$^{-1}$ | Precipitation |
| $Q$ | kg kg$^{-1}$ | Specific humidity |
| $R_s^d(0)$ | W m$^{-2}$ | Downward shortwave radiant flux density at the canopy top |
| $R_l^d(0)$ | W m$^{-2}$ | Downward longwave radiant flux density at the canopy top |
| $T_a$ | K | Air temperature |
| $U$ | m s$^{-1}$ | Wind speed |

**Table 2.** Variables exchanged between the land surface model (LSM) and crop growth model (CGM)

| Variable | Unit | Description |
| --- | --- | --- |
| **LSM to CGM** | | |
| $R_1^d(l)$ | W m$^{-2}$ | direct downward radiant flux density for photosynthesis active radiation (PAR) at a leaf area index (LAI) depth of $l$ |
| $S_1^d(l)$ | W m$^{-2}$ | scattered downward radiant flux density for PAR at a LAI depth of $l$ |
| $S_1^u(l)$ | W m$^{-2}$ | scattered upward radiant flux density for PAR at a LAI depth of $l$ |
| $T_c$ | K | canopy temperature |
| **CGM to LSM** | | |
| $\overline{g}_s$ | m s$^{-1}$ | stomatal conductance per unit leaf area for both sides of the leaf |
| $h_{gt}$ | m | canopy height |
| $L$ | m$^2$ m$^{-2}$ | LAI |
| $W_{sh}$ | kg ha$^{-1}$ | dry matter weight of shoot |
| $z_{rt}$ | m | root depth |





Table 3: Variables

| Symbol | Units | Eq. | Description |
|---|---|---|---|
| $\overline{A}_{g,x}$ | $\mathrm{mol(CO_2)\,m^{-2}(l)s^{-1}}$ | 61 | gross primary production per unit leaf area of sunlit ($\overline{A}_{g,sn}$) and shade($\overline{A}_{g,sh}$) leaves |
| $\overline{A}_{g',x}$ | $\mathrm{mol(CO_2)\,m^{-2}(l)s^{-1}}$ | 65 | gross primary production without photosynthesis down-regulation per unit leaf area of sunlit ($\overline{A}_{g',sn}$) and shade($\overline{A}_{g',sh}$) leaves |
| $A_n$ | $\mathrm{mol(CO_2)\,m^{-2}s^{-1}}$ | 56 | net carbon assimilation |
| $\overline{A}_{n,x}$ | $\mathrm{mol(CO_2)\,m^{-2}(l)\,s^{-1}}$ | 57 | net carbon assimilation per unit leaf area of sunlit ($\overline{A}_{n,sn}$) and shade ($\overline{A}_{n,sh}$) leaves |
| $A_{3,i}$ | - | C8 | variable for the calculation of coefficients of radiation equations (Eqs. C2 and C3) |
| $A^+$ | - | E6 | intermediate variable for the calculation of roughness |
| $C_E$ | - | 28 | BTC for latent heat between the entire surface and atmosphere |
| $C_{Ec}$ | - | 25 | bulk transfer coefficients (BTC) for latent heat between canopy and atmosphere |
| $C_{Ew}$ | - | 24 | BTC for latent heat between surface water and atmosphere |
| $C_{Hc}$ | - | 27 | BTC for sensible heat between canopy and atmosphere |
| $C_{Hw}$ | - | 26 | BTC for sensible heat between surface water and atmosphere |
| $C_M$ | - | 37 | BTC for momentum between the entire surface and atmosphere |
| $C_{Mw}$ | - | 38 | BTC for momentum between surface water and atmosphere |
| $C_{x,i}$ | - | C2 to C5 | coefficients of radiation equations (Eqs. 12–14; $x = 1, 2, 3, 4$) |
| $C_X^0$ | - | E7 | intermediate variable for the calculation of roughness (X denotes "T" or "Q") |
| $C_X^\infty$ | - | E8 | intermediate parameter for the calculation of roughness (X denotes "T" or "Q") |
| $c_a$ | Pa | 89 | partial pressure of atmospheric $CO_2$ |
| $c_e$ | - | E15 | leaf transfer coefficient for specific humidity |
| $c_{hs}(z)$ | $\mathrm{J\,m^{-3}\,K^{-1}}$ | 47 | volumetric heat capacity of soil at a depth of z |
| $c_{i,x}$ | Pa | 57 to 98 | partial pressure of intercellular $CO_2$ |
| $c_{s,x}$ | Pa | 88 | partial pressure of $CO_2$ at leaf boundary |
| $D_i^d(l)$ | $\mathrm{W\,m^{-2}}$ | 12 | radiant flux density for downward direct radiation for photosynthesis active radiation (PAR) ($i = 1$) or near infrared radiation (NIR) ($i = 2$) at a leaf area index (LAI) depth of $l$ |
| $D_g$ | $\mathrm{kg\,m^{-2}\,s^{-1}}$ | 43 | amount of water that falls from canopy onto surface water due to gravity |
| $D_{oy}$ | day | - | the number of days from Jan 1 |
| $DVR$ | K | 103 | development rate at $t$ |
| $DVS$ | - | 101 | development stage at $t$ |
| $d$ | m | E1 | zero-plane displacement height |
| $E_c$ | $\mathrm{kg\,m^{-2}\,s^{-1}}$ | 7 | evaporation from canopy |
| $E_t$ | $\mathrm{kg\,m^{-2}\,s^{-1}}$ | 8 | transpiration from canopy |
| $E_w$ | $\mathrm{kg\,m^{-2}\,s^{-1}}$ | 9 | evaporation from surface water |
| $e_a$ | $\mathrm{P}_a$ | 96 | atmospheric vapour pressure |
| $e_i$ | $\mathrm{P}_a$ | 97 | vapour pressure in leaf |
| $e_{sat}$ | $\mathrm{P}_a$ | 98 | saturated vapour pressure |
| $e_{s,x}$ | $\mathrm{P}_a$ | 94 | vapour pressure at leaf boundary in sunlit ($e_{s,sn}$) and shade ($e_{s,sh}$) leaves |
| $F_s(z)$ | $\mathrm{m^3\,m^{-2}\,s^{-1}}$ | 52 | water flux at a soil depth of z |
| $F_X$ | - | E9 | intermediate parameter for the calculation of roughness (X denotes "T" or "Q") |
| $f_{cw}$ | - | 39 | fraction of canopy that is wet |
| $f_{df}$ | - | 17 | fraction of scattered radiation |
| $f_{dwn}$ | - | 62 | factor of photosynthesis down-regulation |
| $f_{int}$ | - | 42 | interception efficiency of precipitation by canopy |
| $GDS$ | $\mathrm{K \cdot s}$ | 102 | growing degree seconds at $t_s$ |
| $G_{P,glu}$ | $\mathrm{kg\,ha^{-1}\,s^{-1}}$ | 118 and 120 | glucose partitioned to each organ |
| $G_{R,glu}$ | $\mathrm{kg\,ha^{-1}\,s^{-1}}$ | 118 and 120 | growth rate of glucose reserves in leaves |
| $G_{R,pnc}$ | $\mathrm{kg\,ha^{-1}\,s^{-1}}$ | 115 and 119 | growth rate of dry weight for panicles |



*continued*

| Symbol | Units | Eq. | Description |
|---|---|---|---|
| $G_{R,rot}$ | kg ha$^{-1}$ s$^{-1}$ | 116 and 119 | growth rate of dry weight for roots |
| $G_{R,lef}$ | kg ha$^{-1}$ s$^{-1}$ | 113 and 119 | growth rate of dry weight for leaves |
| $G_{R,stc}$ | kg ha$^{-1}$ s$^{-1}$ | 117 and 119 | growth rate of dry weight for starch reserves in stems |
| $G_{R,stm}$ | kg ha$^{-1}$ s$^{-1}$ | 114 and 119 | growth rate of dry weight for stems |
| $G_s(z)$ | W m$^{-2}$ | 46 | heat flux at soil depth of z |
| $G_{ws}$ | W m$^{-2}$ | 10 | heat flux from surface water to soil |
| $\overline{g}_a$ | m s$^{-1}$ | 91 | leaf boundary conductance per unit leaf area for both sides of the leaf |
| $\overline{g}_l$ | mol m$^{-2}(l)$ s$^{-1}$ | 90 | leaf boundary conductance for vapour per unit leaf area |
| $\overline{g}_s$ | m s$^{-1}$ | 99 | stomatal conductance per unit leaf area for both sides of the leaf |
| $\overline{g}_{st}$ | mol m$^{-2}(l)$ s$^{-1}$ | 100 | stomatal conductance for vapour per unit leaf area for both sides of the leaf |
| $\overline{g}_{st,x}$ | mol m$^{-2}(l)$ s$^{-1}$ | 92 | stomatal conductance for vapour per unit leaf area in sunlit ($\overline{g}_{st,sn}$) and shade ($\overline{g}_{st,sh}$) leaves |
| $H_c$ | W m$^{-2}$ | 5 | sensible heat flux from canopy |
| $H_w$ | W m$^{-2}$ | 6 | sensible heat flux from surface water |
| $h_{gt}$ | m | 130 | canopy height |
| $h_{arg}$ | rad | B3 | hour angle from noon ($h_r = 12$) |
| $h_r$ | hour | - | local time at the simulation site |
| $h_{s,x}$ | P$_a$ P$_a^{-1}$ | 93 | relative humidity at leaf boundary in sunlit ($h_{s,sn}$) and shade ($h_{s,sh}$) leaves |
| $I_c$ | kg m$^{-2}$ s$^{-1}$ | 41 | amount of precipitation intercepted by canopy |
| $K(z)$ | kg s m$^{-3}$ | 53 | hydraulic conductivity at a soil depth of z |
| K$_c$ | Pa | 76 | Michaelis constant for CO$_2$ fixation |
| $K_e(z)$ | - | 49 | the Kersten number |
| K$_O$ | Pa | 77 | Michaelis constant for O$_2$ inhibition |
| $k_{ts}(z)$ | W m$^{-1}$ K$^{-1}$ | 48 | thermal conductivity at a soil depth of z |
| $L$ | m$^2$ m$^{-2}$ | 128 | LAI |
| $L_{MO}$ | m | 34 | Monin-Obukhov length of the entire surface |
| $L_{MOw}$ | m | 35 | Monin-Obukhow length of surface water |
| $L_{S,lef}$ | kg ha$^{-1}$ s$^{-1}$ | 124 | loss rate of dry weight for leaves |
| $L_{sn}$ | m$^2(l)$ m$^{-2}$ | 58 | LAI for sunlit leaves |
| $L_{sh}$ | m$^2(l)$ m$^{-2}$ | 59 | LAI for shade leaves |
| $l$ | m$^2(l)$ m$^{-2}$ | - | LAI depth from the top of canopy |
| $P_{R,sh}$ | - | 121 | ratio of glucose partitioned to shoot |
| $P_{R,pnc}$ | - | 123 | ratio of glucose partitioned to panicle from the glucose partitioned to shoot |
| $P_{R,lef}$ | - | 122 | ratio of glucose partitioned to leaf from the glucose partitioned to shoot |
| $P_{1*}$ | - | E11 | intermediate variable for the calculation of roughness (* denotes "M", "T", or "Q") |
| $P_{2*}$ | - | E12 | intermediate variable for the calculation of roughness (* denotes "M", "T", or "Q") |
| $P_{3X}$ | - | E13 | intermediate parameter for the calculation of roughness (X denotes "T" or "Q") |
| $P_{4X}$ | - | E13 | intermediate parameter for the calculation of roughness (X denotes "T" or "Q") |
| $Q_{sat}$ | Kg Kg$^{-1}$ | A2 | specific humidity at saturation |
| $Q_{sn}$ | mol m$^{-2}$ s$^{-1}$ | 80 | photon flux density for PAR absorbed by canopy in sunlit leaves |
| $Q_{sn,d}$ | mol m$^{-2}$ s$^{-1}$ | 82 | direct PAR absorbed in sunlit leaves |
| $Q_{sn,s}$ | mol m$^{-2}$ s$^{-1}$ | 83 | scattered PAR absorbed in shade leaves |
| $Q_{sh}$ | mol m$^{-2}$ s$^{-1}$ | 81 | photon flux density for PAR absorbed by canopy in shade leaves |
| $Q_{sh,s}$ | mol m$^{-2}$ s$^{-1}$ | 84 | scattered PAR absorbed in shade leaves |
| $\overline{Q}_x$ | mol m$^{-2}(l)$ s$^{-1}$ | 79 | photon flux density for PAR absorbed by leaves in sunlit ($\overline{Q}_{sn}$) and shade ($\overline{Q}_{sh}$) leaves |
| $\overline{R}_{d,x}$ | mol(CO$_2$) m$^{-2}(l)$ s$^{-1}$ | 85 | respiration in sunlit ($\overline{R}_{d,sn}$) and shade ($\overline{R}_{d,sh}$) leaves |
| $R_{ex}$ | W m$^{-2}$ | 19 | extraterrestrial radiation |
| $R_{M,stc}$ | kg ha$^{-1}$ s$^{-1}$ | 125 | remobilization rate of dry weight from starch reserves |
| $R_{nc}$ | W m$^{-2}$ | 3 | net radiant flux density at canopy |



*continued*

| Symbol | Units | Eq. | Description |
|---|---|---|---|
| $R_{nw}$ | $\mathrm{W\,m^{-2}}$ | 4 | net radiant flux density at surface water |
| $R_l^d(l)$ | $\mathrm{W\,m^{-2}}$ | 21 | radiant flux density for downward longwave at a LAI depth of $l$ |
| $R_s^d(l)$ | $\mathrm{W\,m^{-2}}$ | 21 | radiant flux density for downward shortwave at a LAI depth of $l$ |
| $R_s^u(l)$ | $\mathrm{W\,m^{-2}}$ | 21 | radiant flux density for upward shortwave at a LAI depth of $l$ |
| $r_{dd,lef}$ | $\mathrm{s^{-1}}$ | 126 | ratio of dead leaf |
| $r_{ij}$ | - | D1 and D2 | reflectivity of canopies ($i=1$:PAR; $i=2$:NIR; $j=1$:direct; $j=2$:scattered) |
| $S$ | - | 78 | Ratio of RuBP partitioned to carboxylase or oxygenase |
| $S_i^d(l)$ | $\mathrm{W\,m^{-2}}$ | 13 | radiant flux density for downward scattered radiation for PAR($i=1$) or NIR ($i=2$) at a LAI depth of $l$ |
| $S_i^u(l)$ | $\mathrm{W\,m^{-2}}$ | 14 | radiant flux density for upward scattered radiation for PAR($i=1$) or NIR ($i=2$) at a LAI depth of $l$ |
| $S_{glu}$ | $\mathrm{kg\,ha^{-1}\,s^{-1}}$ | 110 | supply of glucose to the reserves in leaf |
| $SLW$ | $\mathrm{kg\,m^{-2}}(l)$ | 129 | specific leaf area |
| $S_s(z)$ | $\mathrm{m^3\,m^{-3}\,s^{-1}}$ | 55 | absorption for transpiration by root at soil depth of z |
| $S_{tw}$ | $\mathrm{W\,m^{-2}}$ | 11 | heat flux stored in surface water |
| $T_c$ | K | 3 to 11 | canopy temperature |
| $T_s(z)$ | K | 45 | soil temperature at z of soil depth |
| $T_x$ | K | A2 | temperature of canopy ($T_c$) or surface water ($T_w$) |
| $T_w$ | K | 3 to 11 | surface water temperature |
| $t$ | s | - | time |
| $t_e$ | s | - | time at emergence after sowing |
| $U_c$ | $\mathrm{m\,s^{-1}}$ | F1 | wind speed in the canopy |
| $U_h$ | $\mathrm{m\,s^{-1}}$ | F2 | reference wind speed |
| $V_{max}(l)$ | $\mathrm{mol(CO_2)\,m^{-2}}(l)\,\mathrm{s^{-1}}$ | 74 | reference value for maximum Rubisco capacity at a LAI depth of $l$ |
| $\overline{V}_{max,x}$ | $\mathrm{mol(CO_2)\,m^{-2}}(l)\,\mathrm{s^{-1}}$ | 72 and 73 | reference value for maximum Rubisco capacity per unit leaf area of sunlit and shade leaves |
| $\overline{V}_{mc,x}$ | $\mathrm{mol(CO_2)\,m^{-2}}(l)\,\mathrm{s^{-1}}$ | 69 | maximum Rubisco capacity per unit leaf area of sunlit ($\overline{V}_{mc,sn}$) and shade ($\overline{V}_{mc,sh}$) leaves for $\overline{\omega}_{c,x}$ |
| $\overline{V}_{ms,x}$ | $\mathrm{mol(CO_2)\,m^{-2}}(l)\,\mathrm{s^{-1}}$ | 70 | maximum Rubisco capacity per unit leaf area of sunlit ($\overline{V}_{ms,sn}$) and shade ($\overline{V}_{ms,sh}$) leaves for $\overline{\omega}_{s,x}$ |
| $W_{glu}$ | $\mathrm{kg\,ha^{-1}}$ | 109 | dry weight of glucose reserves in leaves |
| $W_{pnc}$ | $\mathrm{kg\,ha^{-1}}$ | 106 | dry weight of panicles |
| $W_{pnc,mt}$ | $\mathrm{kg\,ha^{-1}}$ | - | dry weight of panicles at maturity |
| $W_{rot}$ | $\mathrm{kg\,ha^{-1}}$ | 107 | dry weight of roots |
| $W_{sh}$ | $\mathrm{kg\,ha^{-1}}$ | 127 | dry weight of shoot |
| $W_{stc}$ | $\mathrm{kg\,ha^{-1}}$ | 108 | dry weight of starch reserves in stems |
| $W_{stm}$ | $\mathrm{kg\,ha^{-1}}$ | 105 | dry weight of stems |
| $w_c$ | m | 40 | amount of water stored in canopy |
| $w_{cap}$ | m | 44 | canopy water capacity |
| $w_s(z)$ | $\mathrm{m^3\,m^{-3}}$ | 50 and 51 | volumetric concentration of soil water at a soil depth of z |
| $Yld$ | $\mathrm{kg\,ha^{-1}}$ | 132 | crop yield |
| $z$ | m | - | soil depth |
| $z_M$ | m | E2 | roughness length of the entire surface for momentum profile |
| $z_{Mw}$ | m | E4 | roughness length that express the effect of water surface on the profile of momentum |
| $z_M^+$ | m | E10 | intermediate variable for the calculation of roughness |
| $z_Q$ | m | E3 | roughness length of the entire surface for specific humidity profile |
| $z_{Qw}$ | m | E5 | roughness length that express the effect of water surface on the profile of specific humidity |
| $z_{rt}$ | m | 131 | root depth |



*continued*

| Symbol | Units | Eq. | Description |
|---|---|---|---|
| $z_T$ | m | E3 | roughness length of the entire surface for temperature profile |
| $z_{Tw}$ | m | E5 | roughness length that express the effect of water surface on the profile of temperature |
| $z_X^+$ | m | E10 | intermediate variable for the calculation of roughness (X denotes "T" or "Q") |
| $z_*^+$ | m | E10 | intermediate variable for the calculation of roughness (* denotes "M", "T", or "Q") |
| $\delta_s$ | rad | B2 | declination of the sun |
| $\Gamma^*$ | Pa | 75 | light compensation point |
| $\gamma_m$ | - | F3 | coefficient of exponential decrease for wind speed in the canopy |
| $\overline{\omega}_{c,x}$ | $\mathrm{mol(CO_2)\,m^{-2}(l)\,s^{-1}}$ | 66 | Rubisco limited assimilation in sunlit ($\overline{\omega}_{c,sn}$) and shade ($\overline{\omega}_{c,sh}$) leaves |
| $\overline{\omega}_{e,x}$ | $\mathrm{mol(CO_2)\,m^{-2}(l)\,s^{-1}}$ | 67 | light-limited assimilation in sunlit ($\overline{\omega}_{e,sn}$) and shade ($\overline{\omega}_{e,sh}$) leaves |
| $\overline{\omega}_{p,x}$ | $\mathrm{mol(CO_2)\,m^{-2}(l)\,s^{-1}}$ | 64 | Rubisco and light-limited assimilation in sunlit ($\overline{\omega}_{p,sn}$) and shade ($\overline{\omega}_{p,sh}$) leaves |
| $\overline{\omega}_{s,x}$ | $\mathrm{mol(CO_2)\,m^{-2}(l)\,s^{-1}}$ | 68 | sucrose limited assimilation for sunlit ($\overline{\omega}_{s,sn}$) and shade ($\overline{\omega}_{s,sh}$) leaves |
| $\Psi_E$ | - | 31 | diabatic correction factor for vapour |
| $\Psi_H$ | - | 31 | diabatic correction factor for heat |
| $\Psi_M$ | - | 30 | diabatic correction factor for momentum |
| $\psi(z)$ | $\mathrm{J\,kg^{-1}}$ | 54 | water potential at a soil depth of z |
| $\rho_a$ | $\mathrm{kg\,m^{-3}}$ | A1 | air density |
| $\tau_{atm}$ | - | 18 | transmissivity of atmosphere |
| $\tau_{cs}$ | - | 20 | transmissivity of canopy for shortwave radiation |
| $\tau_{cl}$ | - | 23 | transmissivity of canopy for longwave radiation |
| $\tau_{ij}$ | - | D3 and D4 | transmissivity of canopy ($i = 1$:PAR; $i = 2$:NIR; $j = 1$:direct; $j = 2$:scattered) |
| $\Theta_0$ | K | 36 | potential temperature |
| $\theta$ | rad | B1 | zenith angle of the sun |
| $\zeta$ | - | 32 | atmospheric stability between the entire canopy and atmosphere |
| $\zeta_w$ | - | 33 | atmospheric stability between surface water and atmosphere |



**Table 4.** Physical and chemical constants

| Variable | Value | Units | Description |
| --- | --- | --- | --- |
| $C_{CO_2,glu}$ | $1.08*10^6$ | kg ha$^{-1}$ h$^{-1}$ /(mol m$^{-2}$ s$^{-1}$) | conversion factor from $CO_2$ to glucose |
| $C_{glu,stc}$ | 0.9 | kg ha$^{-1}$/(kg ha$^{-1}$) | conversion factor of dry weight from glucose to starch |
| $C_{stc,glu}$ | 1.11 | kg ha$^{-1}$/(kg ha$^{-1}$) | conversion factor of dry weight from starch to glucose |
| $c_{pa}$ | 1004.6 | J K$^{-1}$ Kg$^{-1}$ | specific heat of air |
| $c_{pw}$ | 4200 | J K$^{-1}$ Kg$^{-1}$ | specific heat of water |
| $g$ | 9.8 | m s$^{-1}$ | gravitational constant |
| $e_{sat}(T_0)$ | 611 | Pa | vapour pressure at melting temperature of water |
| $k_q$ | $4.6*10^{-6}$ | (mol m$^{-2}$ s$^{-1}$ ) /(W m$^{-2}$) | transfer constant from radiant flux density to photon flux density |
| $k_w$ | 0.6 | W m$^{-1}$ K$^{-1}$ | thermal conductivity of water |
| $R_{dry}$ | 287.04 | J kg$^{-1}$ K$^{-1}$ | gas constant of dry air |
| $R_{sun}$ | 1370 | W m$^{-2}$ | solar constant |
| $R_{vap}$ | 461 | J kg$^{-1}$ K$^{-1}$ | gas constant of vapour |
| $T_0$ | 273.15 | K | melting temperature of water |
| $w_{H_2O}$ | 0.018 | kg/mol | molar weight of vapour |
| $\kappa$ | 0.4 | - | Karman constant |
| $\lambda$ | $2.5*10^6$ | J kg$^{-1}$ | latent heat of vaporisation |
| $\rho_w$ | 1000 | kg m$^{-3}$ | water density |
| $\sigma$ | $5.67*10^{-8}$ | W m$^{-2}$ K$^{-4}$ | Boltzmann constant |





Table 5: Parameters

| Variable | Value | Units | Description | Source |
|---|---|---|---|---|
| **Simulation setting** | | | | |
| $C_a$ | - | ppm | atmospheric $CO_2$ concentration | Masutomi et al. (2016) |
| $d_w$ | - | m | depth of surface water | Masutomi et al. (2016) |
| $L_t$ | - | degree | latitude of the simulation site | Masutomi et al. (2016) |
| $Sw_{DOY}$ | - | DOY | DOY of sowing day | Masutomi et al. (2016) |
| $W_{glu,0}$ | - | kg/ha | dry weight of glucose reserve at emergence | Masutomi et al. (2016) |
| $W_{lef,0}$ | - | kg/ha | dry weight of leaf at emergence | Masutomi et al. (2016) |
| $W_{rot,0}$ | - | kg/ha | dry weight of root at emergence | Masutomi et al. (2016) |
| $W_{stm,0}$ | - | kg/ha | dry weight of stem at emergence | Masutomi et al. (2016) |
| $z_a$ | - | m | reference height at which wind speed is observed | Masutomi et al. (2016) |
| $z_{max}$ | - | m | depth of soil layer | Masutomi et al. (2016) |
| $z_{sat}$ | - | m | depth to which soil is saturated | Masutomi et al. (2016) |
| $z_b$ | - | m | depth from the soil surface to the upper bound of the bottommost layer of soil | Masutomi et al. (2016) |
| $\delta t$ | - | s | time resolution | Masutomi et al. (2016) |
| **Soil-type specific** | | | | |
| $B$ | - | - | factor for hydraulic conductivity and water potential | Masutomi et al. (2016) |
| $K_s$ | - | kg s m$^{-3}$ | hydraulic conductivity at saturation | Masutomi et al. (2016) |
| $w_{sat}$ | - | m$^3$ m$^{-3}$ | volumetric concentration of soil water at saturation | Masutomi et al. (2016) |
| $\psi_s$ | - | J kg$^{-1}$ | water potential at saturation | Masutomi et al. (2016) |
| $\rho_s$ | - | kg m$^{-3}$ | soil bulk density | Masutomi et al. (2016) |
| **Crop-specific (paddy rice)** | | | | |
| $b$ | 0.01 | mol m$^{-2}$ s$^{-1}$ | intercept of the Ball-Berry model | Sellers et al. (1996b) |
| $C_{glu,lef}$ | 0.955 | kg ha$^{-1}$/(kg ha$^{-1}$) | conversion factor of dry weight from glucose to leaf | Penning de Vries et al. (1989) |
| $C_{glu,pnc}$ | 0.821 | kg ha$^{-1}$/(kg ha$^{-1}$) | conversion factor of dry weight from glucose to panicle | Penning de Vries et al. (1989) |
| $C_{glu,rot}$ | 0.928 | kg ha$^{-1}$/(kg ha$^{-1}$) | conversion factor of dry weight from glucose to root | Penning de Vries et al. (1989) |
| $C_{glu,stm}$ | 0.928 | kg ha$^{-1}$/(kg ha$^{-1}$) | conversion factor of dry weight from glucose to stem | Penning de Vries et al. (1989) |
| $c_h$ | 0.06 | - | leaf transfer coefficient for heat | Kimura and Kondo (1998) |
| $c_m$ | 0.2 | - | leaf transfer coefficient for momentum | Kimura and Kondo (1998) |
| $DVS_{rot1}$ | Parameterized | - | 1st point of DVS at which the partition to root changes | Masutomi et al. (2016) |
| $DVS_{rot2}$ | Parameterized | - | 2nd point of DVS at which the partition to root changes | Masutomi et al. (2016) |
| $DVS_{lef1}$ | Parameterized | - | 1st point of DVS at which the partition to leaf changes | Masutomi et al. (2016) |
| $DVS_{lef2}$ | Parameterized | - | 2nd point of DVS at which the partition to leaf changes | Masutomi et al. (2016) |
| $DVS_{pnc1}$ | Parameterized | - | 1st point of DVS at which the partition to panicle changes | Masutomi et al. (2016) |
| $DVS_{pnc2}$ | Parameterized | - | 2nd point of DVS at which the partition to panicle changes | Masutomi et al. (2016) |
| $eDVS$ | Parameterized | - | DVS at emergence | Masutomi et al. (2016) |
| $f_d$ | 0.015 | - | respiration factor | Sellers et al. (1996b) |
| $f_{stc}$ | Parameterized | - | fraction of glucose allocated to starch reserves | Masutomi et al. (2016) |
| $h_{gt,aa}$ | Parameterized | - | parameter for relation between leaf area index (LAI) and height before heading | Masutomi et al. (2016) |
| $h_{gt,ab}$ | Parameterized | - | parameter for relation between LAI and height before heading | Masutomi et al. (2016) |
| $h_{gt,ba}$ | Parameterized | - | parameter for relation between LAI and height after heading | Masutomi et al. (2016) |
| $h_{gt,bb}$ | Parameterized | - | parameter for relation between LAI and height after heading | Masutomi et al. (2016) |
| $hDVS$ | Parameterized | - | DVS at heading | Masutomi et al. (2016) |
| $k_{yld}$ | Parameterized | - | ratio of crop yield to dry weight of panicle at maturity | Masutomi et al. (2016) |
| $k_{SLW}$ | Parameterized | - | parameter for the relation between $SLW$ and $DVS$ | Masutomi et al. (2016) |
| $m$ | 9 | - | the slope of the Ball-Berry model | Sellers et al. (1996b) |
| $mGDS$ | Parameterized | K·s | growing degree second at maturity | Masutomi et al. (2016) |
| $P_{rot}$ | Parameterized | - | ratio of glucose partitioned to root | Masutomi et al. (2016) |
| $P_{lef}$ | Parameterized | - | ratio of glucose partitioned to leaf from glucose partitioned to shoot | Masutomi et al. (2016) |
| $r_{d1,lef}$ | Parameterized | s$^{-1}$ | ratio of dead leaf at harvest | Masutomi et al. (2016) |





*continued* (Table 5)

| Variable | Value | Units | Description | Source |
|---|---|---|---|---|
| $r_{rm,stc}$ | $1.16*10^{-6}$ | $\text{s}^{-1}$ | ratio of remobilization | Bouman et al. (2001) |
| $r_{rt}$ | $1.16*10^{-7}$ | $\text{m s}^{-1}$ | growth ratio of root | Penning de Vries et al. (1989) |
| $r_1$ | 0.105 | - | leaf reflectivity for photosynthesis active radiation (PAR) | Sellers et al. (1996b) |
| $r_2$ | 0.58 | - | leaf reflectivity for near infrared radiation (NIR) | Sellers et al. (1996b) |
| $SLW_{mx}$ | Parameterized | $\text{kg m}^{-2}$ | maximum specific leaf area | Masutomi et al. (2016) |
| $SLW_{mn}$ | Parameterized | $\text{kg m}^{-2}$ | minimum specific leaf area | Masutomi et al. (2016) |
| $s_1$ | Parameterized | $\text{K}^{-1}$ | temperature dependence of $\overline{V}_{max,x}$ on $\overline{V}_{mc,x}$ | Masutomi et al. (2016) |
| $s_2$ | Parameterized | K | temperature dependence of $\overline{V}_{max,x}$ on $\overline{V}_{mc,x}$ | Masutomi et al. (2016) |
| $s_4$ | 281 | K | temperature dependence of $\overline{V}_{max,x}$ on $\overline{V}_{ms,x}$ | Sellers et al. (1996b) |
| $T_b$ | 281.15 | K | minimum temperature for development | Bouman et al. (2001) |
| $T_o$ | 303.15 | K | optimal temperature for development | Bouman et al. (2001) |
| $T_h$ | 313.15 | K | maximum temperature for development | Bouman et al. (2001) |
| $trDVS$ | Parameterized | - | DVS at transplanting and at which transplanting shock starts | Masutomi et al. (2016) |
| $teDVS$ | Parameterized | - | DVS at which transplanting shock ends | Masutomi et al. (2016) |
| $t_1$ | 0.07 | - | leaf transmissivity for PAR | Sellers et al. (1996b) |
| $t_2$ | 0.25 | - | leaf transmissivity for NIR | Sellers et al. (1996b) |
| $V_{max}(0)$ | Parameterized | $\mu\,\text{mol m}^{-2}\,\text{s}^{-1}$ | maximum Rubisco capacity at the canopy top | Masutomi et al. (2016) |
| $z_{rt,mx}$ | 0.3 | m | maximum root depth | Penning de Vries et al. (1989) |
| $\beta_{ce}$ | 0.98 | - | GPP transition factor | Sellers et al. (1996b) |
| $\epsilon_e$ | 0.08 | $\text{mol mol}^{-1}$ | quantum efficiency | Sellers et al. (1996b) |
| **Others** | | | | |
| $A_{x,i}$ | C6–C7 | - | coefficients of radiation equations (Eqs. 12-14; x=1,2) | Watanabe and Ohtani (1995) |
| $a_i$ | C1 | - | extinction coefficient for scattered radiation | Watanabe and Ohtani (1995) |
| $C_0$ | 288 | ppm | base concentration of CO2 for photosynthesis down-regulation | Arora et al. (2009) |
| $c_{pm}$ | 870 | $\text{J kg}^{-1}\,\text{K}^{-1}$ | specific heat of soil minerals | Campbell and Norman (1998) |
| $D_1$ | $1.14*10^{-11}$ | - | coefficient related to gravitational fall of canopy water | Rutter et al. (1975) |
| $D_2$ | $3.7*10^3$ | - | coefficient related to gravitational fall of canopy water | Rutter et al. (1975) |
| $d_f$ | $\sec(2\pi(53/360))$ | - | scattered factor | Watanabe and Ohtani (1995) |
| $F$ | 0.5 | - | distribution of leaf orientation | Goudriaan and van Laar (1994) |
| $K_n$ | 0.3 | - | vertical distribution of nitrogen | Oleson and Lawrence (2013) |
| $k_{ts0}$ | 0.25 | $\text{W m}^{-1}\,\text{K}^{-1}$ | thermal conductivity of dry soil | Campbell and Norman (1998) |
| $k_{tss}$ | 1.58 | $\text{W m}^{-1}\,\text{K}^{-1}$ | thermal conductivity of saturated soil | Best et al. (2011) |
| $[O_2]$ | 20900 | Pa | partial pressure of intercellular $O_2$ | Collatz et al. (1991) |
| $r_g$ | 0.1 | - | albedo of surface water for shortwave radiation | Maruyama and Kuwagata (2010) |
| $s_3$ | 0.2 | $\text{K}^{-1}$ | temperature dependence of $\overline{V}_{max,x}$ on $\overline{V}_{ms,x}$ | Masutomi et al. (2016) |
| $s_5$ | 1.3 | $\text{K}^{-1}$ | temperature dependence on $\overline{R}_{d,x}$ | Sellers et al. (1996b) |
| $s_6$ | 328 | K | temperature dependence on $\overline{R}_{d,x}$ | Sellers et al. (1996b) |
| $z_{Ms}$ | 0.001 | m | roughness length of surface water for momentum | Kimura and Kondo (1998) |
| $z_{Qs}$ | 0.001 | m | roughness length of surface water for specific humidity | Kimura and Kondo (1998) |
| $z_{Ts}$ | 0.001 | m | roughness length of surface water for heat | Kimura and Kondo (1998) |
| $\beta_{pc}$ | 0.95 | - | GPP transition factor | Sellers et al. (1996b) |
| $\epsilon$ | 0.96 | - | longwave emissivity of surface water | Campbell and Norman (1998) |
| $\gamma_d$ | 0.9 | - | response parameter to elevated $CO_2$ | Arora et al. (2009) |
| $\gamma_{gd}$ | 0.42 | - | response parameter to elevated $CO_2$ | Arora et al. (2009) |
| $\tau_b$ | $8.64*10^6$ | s | recession constant for base water flow (100day) | Hanasaki et al. (2008) |