# Peer review of "A land surface model combined with a crop growth model for paddy rice (MATCRO-Rice Ver. 1) – Part I: Model description"

_Geoscientific Model Development, 2016_

## Referee Comment (RC1) · Anonymous Referee #1 · 29 Mar 2016

Dear Authors,

Please find below my comments on your paper "A land surface model combined with a crop growth model for paddy rice (MATCRO-Rice Ver. 1) – Part I: Model description", which describes de adaptation of a LSM, MATSIRO, embedded in a GCM, MIROC-ESM, to simulate the grow and development of irrigated rice, and by consequence the influences of this on the fluxes (energy, water) between land and atmosphere.

The paper is well written. Rice is indeed an important crop worldwide, and the subject is therefore very relevant. In addition to the impact of crop grow, a range of authors have shown that irrigation might have an important impact on the exchange of water and energy between land and atmosphere (it might be useful to add this in your introduction

as a justification).

However I have the following concerns: - The paper does not provide an evaluation, a parametrisation and, most important, a validation of the developed model: I know there is a paper part II on this, but I think that the paper cannot stand on its own without these (I suggest the combine part I and II into one paper). - It is not always very clear what was already part of MATSIRO and what you have developed. It seems to me that only minor adaptations have been implemented up to pg 15 "4.2 Crop development". In this paper you do not describe again all equations of MATSIRO, as this has already been published (you only have to refer to Takata, 2003). By consequence, you can significantly reduce the length of this paper (reduce section 3 to 1-2 pages and section 4.1 to max. 1pg) and add a section on model parametrisation, evaluation and validation. - Some small modifications have been described in section 3 and section 4.1. The explicit reasons for these adaptations are not provided. The impacts of these modifications on the model simulations are missing. What are the added values of those adaptations? Do those modifications affect significantly your model simulations (compared to the original LSM set-up)? Are the modifications you implemented specific for rice or are they more generally applicable? - You assume that your field is flooded (e.g. soil always at saturation level, etc,...): is this assumption correct for the whole year round or only valid during the growing season or parts of it? If this assumption is not valid for the whole year round, can you use this model to make climate simulations, as you suggest in your conclusion? - As you do not do an evaluation of your model, you cannot write some of the statements in your abstract (e.g. l2:"...accurate simulation...": you don't show this in your paper), ...and conclusion. - Describe explicitly in your paper which variables (of your adapted model), are now exchanged between the LSM and the GCM, as you mention in introduction at l14.

I think that the paper needs some major revisions before publication.

Best Regards,

---

## Referee Comment (RC2) · Anonymous Referee #2 · 1 May 2016

This paper is the description of the model of a land surface model (MATCRO) combined with a crop growth model for paddy rice. The paper well describes the underlying scientific basis. Combining land surface model with crop growth model is an important task to understand the interaction between climate and cropland. The paper also well explains the frameworks of the model. The model description part of this paper is very understandable. The model is designed for consistently simulating latent and sensible heat fluxes, net carbon flux, and crop yield by exchanging variables between the land surface model and crop growth model. The main frame work of the land surface module of this model is the same with original MATCRO, except for the addition of surface water above the soil. The crop growth model module is the combination of several models,

for example, sun-shade model for the assimilation module, a simple canopy model by Watanabe and Ohtani (1995) for calculating PAR, and MACROS Penning de Vries et al., 1989 and ORYZA2000 (Bouman et al., 2001) for crop growth.

I feel that the choices of these modules are appropriate for the purpose of the model. However, the calibrating method of parameters is explained in another submitted paper (Masutomi et al. 2016 in review). The examples of the output of this model are not provided in the current paper and these are provided in the accompanied paper.

It is my understanding that examples of model output should be provided, with evaluation against standard benchmarks, observations in GMD. There appears to be no reason to divide the model description paper to two papers because both the model description part (this paper) and the validation part (another submitted paper) are not so long. However, if dividing the study into two papers is acceptable, I think this paper is acceptable. The model description part is well explained. The equations described in this paper have no mistake.

Specific Comments

p. 5, l. 11: Is this assumption appropriate? The model that the authors are developing is rice specific model. However, the leaf orientation of most Poaceae species would not be random. The required preciseness for the leaf orientation may depend on the purpose of the model (or temporal resolution), but the precise description of the leaf orientation may be needed if the purpose of the model is the estimation of hourly fluctuation of the fluxes. If the purpose of the model is the estimation of crop yield for example, the assumption of the leaf orientation may not have critical effect on the estimation. The authors should add the discussion of the appropriateness of the assumption.

p. 6, l. 19-20: Please explain in detail.

p. 11, l. 28: The down-regulation effect of photosynthesis has a very profound effect

on crop growth. The parameters relevant to photosynthesis down-regulation in Arora et al. (2009) are calculated using mainly plants other than rice. Therefore, the authors should explain the applicability of the parameter values to rice.

Eq. 69-71: Please change the variable name of "Qt". The The character "Q" is already used for the photon flux density.

All equations: Italic should be used for only scalars in principle. For example, it may be preferable not to use Italic for the subscript "c" of "Hc" if "c" is not scalar value. Moreover, upright font (not italic) should be used for multi-letter variables (for example, "Rnc"). Please recheck almost of all subscripts and superscripts of the equations.
* * *

---

## Author Comment (AC1) · 19 May 2016

**Response to Anonymous Referee#1 (gmd-2016-28)**

We will add the explanation on the importance of irrigation in land-atmosphere simulations into the introduction of the revised manuscript, referring some studies, e.g., Boucher et al. (2004), Lobell et al. (2006), Snyder et al. (2006), and Kueppers et al. (2008).
* * *
The paper does not provide an evaluation, a parametrisation and, most important, a validation of the developed model: I know there is a paper part II on this, but I think that the paper cannot stand on its own without these (I suggest the combine part I and II into one paper).
* * *
description and evaluation papers, if the evaluation is extensive. The revised version of the evaluation paper will become more extensive, because we will add the results of two types of simulations into the revised manuscript of the evaluation paper: the effects of model modifications and the validation of the model at the sites which are independent from the parameterization site. Considering the extent of the model evaluation paper, we think the separate submission is acceptable.
* * *
It seems to me that only minor adaptations have been implemented up to pg 15 "4.2 Crop development". In this paper you do not describe again all equations of MATSIRO, as this has already been published (you only have to refer to Takata, 2003). By consequence, you can significantly reduce the length of this paper (reduce section 3 to 1-2 pages and section 4.1 to max. 1pg) and add a section on model parametrisation, evaluation and validation.
* * *
All the modifications except the consideration of water surface are applicable for other crops. We will add this point into the concluding remarks in the revised manuscript.
* * *
You assume that your field is flooded (e.g. soil always at saturation level, etc,...): is this assumption correct for the whole year round or only valid during the growing season or parts of it? If this assumption is not valid for the whole year round, can you use this model to make climate simulations, as you suggest in your conclusion?

The statement in the abstract means "accurate simulations in agricultural land are important for climate simulations". This is a general statement, but might be misleading. We will remove the words "accurate" and "accurately" from the abstract and the introduction. However, we could not find the statements you pointed out in the conclusion. The conclusion just discusses the applicability and limitations of the model.

---

## Author Comment (AC2) · 19 May 2016

Response to Anonymous Referee#2 (gmd-2016-28)

#################################################################

*It is my understanding that examples of model output should be provided, with evaluation against standard benchmarks, observations in GMD. There appears to be no reason to divide the model description paper to two papers because both the model description part (this paper) and the validation part (another submitted paper) are not so long. However, if dividing the study into two papers is acceptable, I think this paper is acceptable.*

#################################################################

Although the present paper does not include the model parameterization and validation, the journal guideline admits the separate submission of model description and evaluation papers, if the evaluation is extensive. The revised version of the evaluation paper will become more extensive, because we will add the results of two types of simulations into the revised manuscript of the evaluation paper: the effects of model modifications and the validation of the model at the sites which are independent from the parameterization site. Considering the extent of the model evaluation paper, we think the separate submission is acceptable.

#################################################################

*p. 5, l. 11: Is this assumption appropriate? The model that the authors are developing is rice specific model. However, the leaf orientation of most Poaceae species would not be random. The required preciseness for the leaf orientation may depend on the purpose of the model (or temporal resolution), but the precise description of the leaf orientation may be needed if the purpose of the model is the estimation of hourly fluctuation of the fluxes. If the purpose of the model is the estimation of crop yield for example, the assumption of the leaf orientation may not have critical effect on the estimation. The authors should add the discussion of the appropriateness of the assumption.*

#################################################################

To be precise, this assumption is not appropriate. The leaf orientation of crops varies with their growing. However, no data is available on the change in the leaf

orientation for rice. Therefore, we assumed that it is random. As you pointed out, the required accuracy depends on the purpose of the model. In the revised manuscript of the evaluation paper, we will add the results of the comparison of hourly fluxes between simulations and observations. The results showed that the simulations are in good agreement with the observations for the hourly fluxes.

We will add the above discussion into the revised manuscript.

###################################################################
*p. 6, l. 19-20: Please explain in detail.*
###################################################################
The equation of the scattered factor, df=sec(2pi*(53/360)), is related to the third assumption in the previous page (P5). The detail of the assumption was explained at P5 L14-15 in the manuscript. To make clear the relation between the equation and assumption, we will add the above explanation in the revised manuscript.

###################################################################
*p. 11, l. 28: The down-regulation effect of photosynthesis has a very profound effect on crop growth. The parameters relevant to photosynthesis down-regulation in Arora et al. (2009) are calculated using mainly plants other than rice. Therefore, the authors should explain the applicability of the parameter values to rice.*
###################################################################
We think the down-regulation effect is limited under the current $CO_2$ concentration, but significant under the future. In the manuscript, we "tentatively" used the mean value for the key parameter in the equation of down-regulation in Arora et al. (2009), because there is no information on the key parameter of the equation for rice, according to our knowledge, and the $CO_2$ effect on crop growth still has a large uncertainty. If the value for the key parameter is quantified in the future, the tentative value should be replaced. We will add the above discussion into the revised manuscript.

\#\#\#\#\#\#\#\#\#\#\#\#\#\#\#\#\#\#\#\#\#\#\#\#\#\#\#\#\#\#\#\#\#\#\#\#\#\#\#\#\#\#\#\#\#\#\#\#\#\#\#\#\#\#\#\#\#\#\#\#\#\#\#\#\#\#\#\#\#

*Eq. 69-71: Please change the variable name of "Qt". The The character "Q" is already used for the photon flux density.*

\#\#\#\#\#\#\#\#\#\#\#\#\#\#\#\#\#\#\#\#\#\#\#\#\#\#\#\#\#\#\#\#\#\#\#\#\#\#\#\#\#\#\#\#\#\#\#\#\#\#\#\#\#\#\#\#\#\#\#\#\#\#\#\#\#\#\#

We will change the symbol in the revised manuscript.

\#\#\#\#\#\#\#\#\#\#\#\#\#\#\#\#\#\#\#\#\#\#\#\#\#\#\#\#\#\#\#\#\#\#\#\#\#\#\#\#\#\#\#\#\#\#\#\#\#\#\#\#\#\#\#\#\#\#\#\#\#\#\#\#\#\#\#\#\#

*All equations: Italic should be used for only scalars in principle. For example, it may be preferable not to use Italic for the subscript "c" of "Hc" if "c" is not scalar value. Moreover, upright font (not italic) should be used for multi-letter variables (for example, "Rnc"). Please recheck almost of all subscripts and superscripts of the equations.*

\#\#\#\#\#\#\#\#\#\#\#\#\#\#\#\#\#\#\#\#\#\#\#\#\#\#\#\#\#\#\#\#\#\#\#\#\#\#\#\#\#\#\#\#\#\#\#\#\#\#\#\#\#\#\#\#\#\#\#\#\#\#\#\#\#\#\#

According to your comments, we will recheck and modify the symbols for variables and parameters.

---

## Author Response (AR1)

**Response to Anonymous Referee#1 (gmd-2016-28)**

###############################################################

*In addition to the impact of crop grow, a range of authors have shown that irrigation might have an important impact on the exchange of water and energy between land and atmosphere (it might be useful to add this in your introduction as a justification).*

###############################################################

You are exactly right. A range of studies have shown the impact of irrigation on the atmosphere.

We added the explanation on the importance of irrigation in land-atmosphere simulations into the introduction of the revised manuscript (P2, L8-L12), referring some studies, e.g., Boucher et al. (2004), Lobell et al. (2006), Snyder et al. (2006), and Kueppers et al. (2008).

###############################################################

*The paper does not provide an evaluation, a parametrisation and, most important, a validation of the developed model: I know there is a paper part II on this, but I think that the paper cannot stand on its own without these (I suggest the combine part I and II into one paper).*

###############################################################

The journal guideline of GMD admits the separate submission of the model description and evaluation papers, if the evaluation is extensive. The revised version of the evaluation paper became more extensive, because we added the results of two types of simulations into the revised manuscript of the evaluation paper: the effects of model modifications and the validation of the model at the sites which are independent from the parameterization site. Considering the extent of the model evaluation paper, we think the separate submission is acceptable.

###############################################################

*It is not always very clear what was already part of MATSIRO and what you have developed.*

###############################################################

To make clear the difference between the original model and our model, we added Table 3, where all the modifications are listed.

###############################################################

It seems to me that only minor adaptations have been implemented up to pg 15 "4.2 Crop development". In this paper you do not describe again all equations of MATSIRO, as this has already been published (you only have to refer to Takata, 2003). By consequence, you can significantly reduce the length of this paper (reduce section 3 to 1-2 pages and section 4.1 to max. 1pg) and add a section on model parametrisation, evaluation and validation.

###############################################################

If we only refer to Takata et al. (2003) and do not show the equations, nobody can develop the model and reproduce the results, because Takata et al. (2003) did not describe all parts of MATSIRO. According to the journal guideline of GMD, the model reproducibility is emphasized. Hence, we showed all the equations, which are necessary for developing the model and reproducing the results. We also note that almost of the equations which we showed in the manuscript were not shown in Takata et al. (2003). We think that only referring to Takata et al. (2003) is not a good option for ensuring the model reproducibility.

###############################################################

*Some small modifications have been described in section 3 and section 4.1. The explicit reasons for these adaptations are not provided. The impacts of these modifications on the model simulations are missing. What are the added values of those adaptations? Do those modifications affect significantly your model simulations (compared to the original LSM set-up)? Are the modifications you implemented specific for rice or are they more generally applicable?*

##################################################################

We absolutely agree with you. Hence, we showed the effects of the modifications to the revised manuscript of the evaluation paper, by comparing the simulations between the original model and our model.

All the modifications except the consideration of water surface are applicable for other crops. We added this point into the section 2 (P2, L31-32).

##################################################################

*You assume that your field is flooded (e.g. soil always at saturation level, etc,...): is this assumption correct for the whole year round or only valid during the growing season or parts of it? If this assumption is not valid for the whole year round, can you use this model to make climate simulations, as you suggest in your conclusion?*

##################################################################

The model described in the previous manuscript can't simulate fluxes for the whole year, because the model focused on only the growing periods of paddy rice. As you pointed out, the model should be able to simulate fluxes for the whole year in order to apply the model to climate simulations. Therefore we improved the model so that it can simulate fluxes for the whole year, even under non-flooded and rainfed condition. To describe the improved model, we drastically modified the model description (throughout the manuscript) and equations (Eqs. 7, 8, 9, 11, 24, 53, 54, 76 and 77) and added new equations (Eqs. 30, 46, 47, 59, 60, 61, 62, 78 and 79). In addition, we validated the simulated LHF and SHF for the whole year in the validation paper.

##################################################################

*As you do not do an evaluation of your model, you cannot write some of the statements in your abstract (e.g. l2:"...accurate simulation...": you don't show this in your paper), ...and conclusion.*

##################################################################

The statement in the abstract means "accurate simulations in agricultural land are

important for climate simulations". This is a general statement, but might be misleading. We removed the words "accurate" and "accurately" from the abstract and the introduction. However, we could not find the statements you pointed out in the conclusion. The conclusion just discusses the applicability and limitations of the model.

############################################################
*Describe explicitly in your paper which variables (of your adapted model), are now exchanged between the LSM and the GCM, as you mention in introduction at l14.*
############################################################
Table 2 shows the variables which are exchanged into the LSM and CGM.

Response to Anonymous Referee#2 (gmd-2016-28)

##################################################################

*It is my understanding that examples of model output should be provided, with evaluation against standard benchmarks, observations in GMD. There appears to be no reason to divide the model description paper to two papers because both the model description part (this paper) and the validation part (another submitted paper) are not so long. However, if dividing the study into two papers is acceptable, I think this paper is acceptable.*

##################################################################

Although the present paper does not include the model parameterization and validation, the journal guideline admits the separate submission of model description and evaluation papers, if the evaluation is extensive. The revised version of the evaluation paper became more extensive, because we added the results of two types of simulations into the revised manuscript of the evaluation paper: the effects of model modifications and the validation of the model at the sites which are independent from the parameterization site. Considering the extent of the model evaluation paper, we think the separate submission is acceptable.

##################################################################

*p. 5, l. 11: Is this assumption appropriate? The model that the authors are developing is rice specific model. However, the leaf orientation of most Poaceae species would not be random. The required preciseness for the leaf orientation may depend on the purpose of the model (or temporal resolution), but the precise description of the leaf orientation may be needed if the purpose of the model is the estimation of hourly fluctuation of the fluxes. If the purpose of the model is the estimation of crop yield for example, the assumption of the leaf orientation may not have critical effect on the estimation. The authors should add the discussion of the appropriateness of the assumption.*

##################################################################

To be precise, this assumption is not appropriate. The leaf orientation of crops varies with their growing. However, no data is available on the change in the leaf

orientation for rice. Therefore, we assumed that it is random. As you pointed out, the required accuracy depends on the purpose of the model. In the revised manuscript of the evaluation paper, we added the results of the comparison of hourly fluxes between simulations and observations. The results showed that the simulations are in good agreement with the observations for the hourly fluxes. We added the above discussion into the revised manuscript (P5, L17-19).

################################################################

*p. 6, l. 19-20: Please explain in detail.*

################################################################

The equation of the scattered factor, $df=\sec(2\pi*(53/360))$, is related to the third assumption shown in P5. The detail of the assumption was explained at P5 L19-20 in the revised manuscript. To make clear the relation between the equation and assumption, we modified the sentence (P6, L25).

################################################################

*p. 11, l. 28: The down-regulation effect of photosynthesis has a very profound effect on crop growth. The parameters relevant to photosynthesis down-regulation in Arora et al. (2009) are calculated using mainly plants other than rice. Therefore, the authors should explain the applicability of the parameter values to rice.*

################################################################

We think the down-regulation effect is limited under the current $CO_2$ concentration, but significant under the future. In the manuscript, we "tentatively" used the mean value for the key parameter in the equation of down-regulation in Arora et al. (2009), because there is no information on the key parameter of the equation for rice, according to our knowledge, and the $CO_2$ effect on crop growth still has a large uncertainty. If the value for the key parameter is quantified in the future, the tentative value should be replaced. We added the above discussion into the revised manuscript (P13, L3 - P13, L7).

###############################################################

*Eq. 69-71: Please change the variable name of "Qt". The The character "Q" is already used for the photon flux density.*

###############################################################

We changed the symbol to "q" in the revised manuscript (P14, Eq. 80).

###############################################################

*All equations: Italic should be used for only scalars in principle. For example, it may be preferable not to use Italic for the subscript "c" of "Hc" if "c" is not scalar value. Moreover, upright font (not italic) should be used for multi-letter variables (for example, "Rnc"). Please recheck almost of all subscripts and superscripts of the equations.*

###############################################################

According to your comments, we modified the symbols for all the variables and parameters (throughout the manuscript).

[revised manuscript text omitted]

$$E_c = \min\{f_{cw}\rho_a C_{Hc} U(Q_{sat}(T_c, P_a) - Q), E_{c,\max}\}, \tag{7}$$

$$E_t =  \begin{cases} \min\{(1-f_{cw})\rho_a C_{Ec}U(Q_{sat}(T_c,P_a)-Q), E_{t,\max}\}, & (\text{if } Q_{sat}(T_c,P_a) > Q) \\ (1-f_{cw})\rho_a C_{Hc}U(Q_{sat}(T_c,P_a)-Q), & (\text{otherwise}) \end{cases} \tag{8}$$

$$E_g =  \begin{cases} \min\{\rho_a C_{Eg}U(h_{ms}Q_{sat}(T_g,P_a)-Q), E_{g,\max}\}, & (\text{if } h_s Q_{sat}(T_g,P_a) > Q) \\ \rho_a C_{Hg}U(h_{ms}Q_{sat}(T_g,P_a)-Q), & (\text{otherwise}) \end{cases} \tag{9}$$

$$G_{gs} = k_w(T_g - T_s(0))/d_w, \tag{10}$$

$$S_{tw} =  \begin{cases} c_{pw}\rho_w d_w(dT_g/dt), & (\text{flooded}), \\ 0 & (\text{unflooded}) \end{cases} \tag{11}$$

where  $R_s^d(0), R_l^d(0), \text{and } R_s^u(0)$ are the downward shortwave radiant flux density, downward longwave radiant flux density, and upward shortwave radiant flux density at the canopy top, respectively,  $\tau_{cs}$ and $\tau_{cl}$ are the canopy transmissivity for shortwave and longwave radiation, respectively,  $C_{Hc}$ and $C_{Hg}$ are the bulk transfer coefficients (BTCs) for sensible heat between canopy and atmosphere and between surface  and atmosphere, respectively,  $C_{Ec}$ and $C_{Eg}$ are the BTCs for latent heat between canopy and atmosphere and between  surface and atmosphere, respectively,  $T_a, P_a$, $U$, and $Q$ are air temperature, air pressure, wind speed, and specific humidity, respectively,  $f_{cw}$ is the fraction of wet canopy,  $h_{ms}$ is humidity of the topsoil, $T_c, T_g, \text{and } T_s(0)$ are the canopy, surface, and topsoil temperature, respectively,  $E_{t,\max}, E_{g,\max}, \text{and } E_{c,\max}$ are the maximum transpiration from canopy, the maximum evaporation from surface and, the maximum evaporation from the canopy, respectively, $c_{pa}$ and $c_{pw}$ are the specific air and water heat, respectively,  $k_w$ is the water thermal conductivity,  $\rho_w$ and $\rho_a$ are water and air density, respectively, $\sigma$ is the Boltzmann constant,  $Q_{sat}$ is specific humidity at saturation,  $d_w$ is the depth of surface water in the case of flooded surface, $\epsilon$ is the longwave emissivity of surface, and $d/dt$ indicates the time differentiation. The argument of the radiant flux density denotes LAI depth from the canopy top, and the argument of soil temperature denotes soil depth from the soil surface. Therefore,  $R_s^d(0), R_l^d(0), \text{and } R_s^u(0)$ indicate the radiant flux density at the canopy top, and  $T_s(0)$ indicates the soil surface temperature.

 $T_a, P_a$, $U$, $Q$,  $R_s^d(0), \text{and } R_l^d(0)$ are meteorological forcing inputs (Table 1).  $R_s^u(0), \tau_{cs}, \tau_{cl}, f_{cw}, h_{ms}, C_{Ec}, C_{Eg}, C_{Hc}, C_{Hg}, T_s(0), E_{t,\max}, E_{g,\max} \text{ and } E_{
[revised manuscript text omitted]

$$\quad C_{\underline{Ew}\,Eg} = \underline{\kappa^2} \left[ \ln\left(\frac{z_a - d}{z_{Mw}}\right) \underline{1/C_{Hg}} + \underline{\Psi_M(\zeta_w)} \underline{r_s U} \right]^{-1} \ln\left(\frac{z_a - d}{z_{Qw}}\right) + \underline{\Psi_E(\zeta_w)}^{-1}, \tag{24}$$

$$C_{\underline{Ec}\,Ec} = C_{\underline{E}E} - C_{\underline{Ew}\,Eg}, \tag{25}$$

$$C_{\underline{Hw}\,Hg} = \kappa^2 \left[ \ln\left(\frac{z_a - d}{z_{Mw}}\right) \ln\left(\frac{z_a - d}{z_{Mg}}\right) + \Psi_{\underline{M}M}(\zeta_{\underline{wg}}) \right]^{-1} \left[ \ln\left(\frac{z_a - d}{z_{Tw}}\right) \ln\left(\frac{z_a - d}{z_{Tg}}\right) + \Psi_{\underline{H}H}(\zeta_{\underline{wg}}) \right]^{-1}, \tag{26}$$

$$C_{\underline{Hc}\,Hc} = C_{\underline{H}H} - C_{\underline{Hw}\,Hg}, \tag{27}$$

where $ C_E \text{ and } C_H$ are the BTCs for latent and sensible heat between the entire surface (canopy + surface)
10 and atmosphere and are given by

$$C_{\underline{E}E} = \kappa^2 \left[ \ln\left(\frac{z_a - d}{z_M}\right) \ln\left(\frac{z_a - d}{z_M}\right) + \Psi_{\underline{M}M}(\zeta) \right]^{-1} \left[ \ln\left(\frac{z_a - d}{z_Q}\right) \ln\left(\frac{z_a - d}{z_Q}\right) + \Psi_{\underline{E}E}(\zeta) \right]^{-1}, \tag{28}$$

$$C_{\underline{H}H} = \kappa^2 \left[ \ln\left(\frac{z_a - d}{z_M}\right) \ln\left(\frac{z_a - d}{z_M}\right) + \Psi_{\underline{M}M}(\zeta) \right]^{-1} \left[ \ln\left(\frac{z_a - d}{z_T}\right) \ln\left(\frac{z_a - d}{z_T}\right) + \Psi_{\underline{H}H}(\zeta) \right]^{-1}. \tag{29}$$

In Eqs. 24 to 29, $\kappa$ is the Karman constant, $d$ is the zero-plane displacement height, $ z_a$ is the reference height at which wind velocity is observed, $ z_{Mg} \text{ and } z_{Tg}$ are the roughness lengths that express the effect of surface  on
15 the profiles of momentum  and temperature, respectively, $z_M, z_T, \text{and } z_Q$ are the roughness lengths of an entire surface (canopy + surface) for the profiles of momentum, temperature, and specific humidity, respectively. , $r_s$ is resistance of topsoil to evaporation. $z_a$ is a simulation setting parameter (Table 6), and $d$, $ z_M, z_T, z_Q, z_{Mg} \text{ and } z_{Tg}$ are the functions of crop height and LAI (Appendix E).  $r_s$ is given by

$$\quad r_s = 800(1 - w_s(0)/w_{sat})/(0.2 + w_s(0)/w_{sat}), \tag{30}$$

where $w_s(0)$ is the water content of topsoil and is calculated in Eq. 53, and $ w_{sat}$ is the soil water content at saturation and is a soil-type specific parameter. $\Psi_M$, $\Psi_H$, and $\Psi_E$ are the diabatic correction factors for momentum, heat, and vapour transport, respectively. The factors are functions of atmospheric stability $\zeta$ as follows:

$$\Psi_{\underline{M}M}(\zeta) = \begin{cases} 6\ln(1+\zeta) & (\zeta > 0 : \text{stable}) \\ -1.2\ln\left[\frac{1+(1-16\zeta)^{1/2}}{2}\right] & (\text{Otherwise: unstable}), \end{cases} \tag{31}$$

$$\Psi_{H}(\zeta) = \Psi_{E}(\zeta) = \begin{cases} 6\ln(1+\zeta) & (\zeta > 0 : \text{stable}) \\ -2\ln\left[\frac{1+(1-16\zeta)^{1/2}}{2}\right] & (\text{Otherwise: stable}). \end{cases} \tag{32}$$

The equations above are adopted from Campbell and Norman (1998), whereas the original MATSRIO model employs different equations. The variable $\zeta$ is replaced by either the atmospheric stability between the entire surface and atmosphere ($\zeta$) or the atmospheric stability between surface and atmosphere ($\zeta_g$). These are given by

$$\zeta = \frac{z_a - d}{L_{MO}}, \tag{33}$$

$$\zeta_g = \frac{z_a - d}{L_{MOg}}, \tag{34}$$

where $L_{MO}$ and $L_{MOg}$ are the Monin-Obukhov lengths for the exchange between the entire surface and atmosphere and between the surface and atmosphere, respectively, and are given by

$$L_{MO} = \frac{\Theta_0 C_M^{3/2} U^2}{\kappa g\{C_{Hg}(T_g - T_a) + C_{Hc}(T_c - T_a)\}}, \tag{35}$$

$$L_{MOg} = \frac{\Theta_0 C_{Mg}^{3/2} U^2}{\kappa g C_{Hg}(T_g - T_a)}, \tag{36}$$

where $g$ is the gravitational constant, $T_g$ and $T_c$ are the temperatures of the surface and canopy, $\Theta_0$ is the potential temperature, $C_M$ and $C_{Mg}$ are the BTC for momentum between an entire surface and atmosphere and between surface and atmosphere, respectively. $C_{Mg}$ in Eq. 36 is introduced according to Maruyama and Kuwagata (2008), while the original MATSIRO uses $C_M$. $T_g$ and $T_c$ are calculated in Section 3.1. $\Theta_0$ is given by

$$\Theta_0 = T_a * (1.0 * 10^5 / P_a)^{(R_{dry}/c_{pa})}, \tag{37}$$

where $R_{dry}$ is the gas constant of dry air. Although the original MATSIRO fixes $\Theta_0$ at 300 K, MATCRO calculates the value according to Campbell and Norman (1998). $C_M$ and $C_{Mg}$ are given by

$$C_M = k^2 \left[\ln\left(\frac{z_a - d}{z_M}\right)\ln\left(\frac{z_a - d}{z_M}\right) + \Psi_M(\zeta)\right]^{-2}, \tag{38}$$

$$C_{Mg} = k^2 \left[\ln\left(\frac{z_a - d}{z_{Mg}}\right)\ln\left(\frac{z_a - d}{z_{Mg}}\right) + \Psi_M(\zeta_g)\right]^{-2}. \tag{39}$$

Now we have six independent equations, Eqs. 24, 25, 26, 27, 38, and 39, for six unknown variables, $C_{Eg}, C_{Ec}, C_{Hg}, C_{Hc}, C_M$, and $C_{Mg}$, respectively. Therefore, we can determine the values of these variables by numerically solving Eqs. 24 to 39. The numerical method is described in Masutomi et al. (2016).

**3.4 Canopy water balance**

The main purpose of this module is to calculate the fraction of wet canopy ($f_{\mathrm{cw}}$) which is used for simulating energy balance at canopy (Section 3.1). To calculate $f_{\mathrm{cw}}$, this module calculates water balance at canopy. Although the module is based on the original MATSIRO, the amount of water that canopies can hold was replaced by using the method described in Penning de Vries et al. (1989). The variable $f_{\mathrm{cw}}$ is given as

$$f_{\mathrm{cw}} = w_{\mathrm{c}}/w_{\mathrm{cap}}, \tag{40}$$

where $w_{\mathrm{c}}$ is the amount of water stored in canopy and $w_{\mathrm{cap}}$ is the water capacity of the canopy. The $w_{\mathrm{c}}$ is calculated by solving the canopy water balance, which is given by

$$\rho_{\mathrm{w}}\frac{dw_{\mathrm{c}}}{dt} = I_{\mathrm{c}} - D_{\mathrm{g}} - E_{\mathrm{c}}, \tag{41}$$

where $\rho_{\mathrm{w}}$ is the density of water, $I_{\mathrm{c}}$ is the amount of precipitation intercepted by canopy, $D_{\mathrm{g}}$ is the amount of water that falls from the canopy onto surface due to gravity, and $E_{\mathrm{c}}$ is the amount of water that evaporates from the canopy (Eq. 7). $I_{\mathrm{c}}$ depends on the amount of precipitation ($P_{\mathrm{r}}$) and LAI ($L$) and is given by

$$I_{\mathrm{c}} = f_{\mathrm{int}}P_{\mathrm{r}}, \tag{42}$$

$$f_{\mathrm{int}} = \begin{cases} L & (L < 1) \\ 1 & (\text{otherwise}) \end{cases}, \tag{43}$$

where $f_{\mathrm{int}}$ indicates the interception efficiency of precipitation by canopy. According to Rutter et al. (1975) and Penning de Vries et al. (1989), $D_{\mathrm{g}}$ and $w_{\mathrm{cap}}$ are given as

$$D_{\mathrm{g}} = \rho_{\mathrm{w}}D_1\exp(D_2 w_{\mathrm{c}}), \tag{44}$$

$$w_{\mathrm{cap}} = (W_{\mathrm{sh}} * 10^{-4})/\rho_{\mathrm{w}}, \tag{45}$$

respectively, where $D_1$ and $D_2$ are parameters (Rutter et al., 1975), and $W_{\mathrm{sh}}$ is the shoot dry weight, which is calculated in the CGM (Eq. 136). In the case of non-flooded surface, the amount of water that falls from the canopy onto soil surface, $F_{\mathrm{c}}$, is calculated by

$$F_{\mathrm{c}} = D_{\mathrm{g}} + (1 - f_{\mathrm{int}})P_{\mathrm{r}} + \max\{0, w_{\mathrm{c}} - w_{\mathrm{cap}}\}\rho_{\mathrm{w}}/\delta t, \quad (\text{unflooded}) \tag{46}$$

where $\delta t$ is the time resolution of simulations. In the case of flooded surface, $F_{\mathrm{c}}$ is not calculated because surface water is present. The maximum evaporation from the canopy ($E_{\mathrm{c,max}}$) is given by

$$E_{\mathrm{c,max}} = w_{\mathrm{c}}\rho_{\mathrm{w}}/\delta t. \tag{47}$$

**3.5  Soil water and heat transfer**

This module calculates heat and water transfer in soil. The main role of this module is to determine the temperature at a soil surface ($T_s(0)$), which is used for simulating energy balance of the surface (Section 3.1). Although this module is based on the original MATSIRO, the calculations of the surface and base runoffs are simplified because hydrological calculations are not the main purpose of MATCRO-Rice.

Soil temperature at a soil depth of $z$ from the soil surface ($T_s(z)$) is calculated from the gradient of heat flux in the soil as follows:

$$c_{hs}(z)\frac{\partial T_s(z)}{\partial t} = \frac{\partial G_s(z)}{\partial z}, \tag{48}$$

where $c_{hs}$ is the volumetric heat capacity of the soil and $G_s(z)$ is the heat flux at a soil depth of $z$ and is given from the gradient of soil temperature

$$G_s(z) = \begin{cases} k_{ts}(z)\frac{\partial T_s(z)}{\partial z} & (0 \le z < z_{max}) \\ 0 & (z = z_{max}). \end{cases} \tag{49}$$

Here, $k_{ts}$ is the soil thermal conductivity. In Eq. 49, we assumed that heat flux at the bottom of the soil layer ($z = z_{max}$) is zero. $z_{max}$ is a simulation setting parameter. When solving Eqs. 48 and 49, the heat flux from surface to soil ($G_{gs}$), calculated in Eq. 10, is used as a boundary condition. The parameter $c_{hs}$ is calculated from the heat capacities of soil components as follows.

$$c_{hs}(z) = \rho_s c_{pm} + \rho_w c_{pw} w_s(z), \tag{50}$$

where $\rho_s$ is the bulk density of soil, $c_{pm}$ is the specific heat of soil minerals, and $w_s(z)$ is the volumetric concentration of soil water. $\rho_s$ is a soil-type specific parameter determined by soil type at a simulation site, and $c_{pm}$ is given according to Campbell and Norman (1998). We note that the first term of the right hand side in Eq. 50 indicates the heat capacity of dry soil. Although the original MATSRIO model assigns a default value to the heat capacity of dry soil for all soil types, MATCRO-Rice calculates the value of the heat capacity of dry soil using the bulk density of soil and the heat capacity of soil minerals, as shown in the first term of Eq. 50. It should be noted that the effect of soil organic matter on $c_{hs}$ is not considered in MATCRO. The parameter $k_{ts}(z)$ in Eq. 49 is given by

$$k_{ts}(z) = K_e(z)(k_{tss} - k_{ts0}) + k_{ts0}, \tag{51}$$

$$K_e(z) = \begin{cases} \log(w_s(z)/w_{sat}) + 1.0 & (\text{if } w_s(z)/w_{sat} \ge 0), \\ 0 & (\text{otherwise}) \end{cases} \tag{52}$$

where $k_{ts0}$ and $k_{tss}$ are the thermal conductivity of dry and saturated soils, respectively, $K_e$ is the Kersten number, and $w_{sat}$ is the volumetric soil water concentration at saturation. $k_{ts0}$ and $k_{tss}$ are parameters. We set $k_{ts0}$=0.25 (Campbell and Norman, 1998), and $k_{tss}$ = 1.58 (Best et al., 2011). The parameter $w_{sat}$ is specific to

soil type. Equations 51 and 52 for the calculation of  $k_{\text{ts}}(z)$ are based on the equations developed by Best et al. (2011), while the original MATSIRO employs a different equation. The variable  $w_{\text{s}}(z)$ depends on the gradient of water flux and absorption by roots at a soil depth $z$ . In addition, water flux from the canopy layer is added into the top layer of the soil ($0 \leq z < z_{\text{t}}$) in the case of non-flooded surface. The variable $w_{\text{s}}(z)$ is given by

$$\quad \underline{w_s(z)}\frac{\partial w_{\text{s}}(z)}{\partial t} = \underline{w_{sat}} \quad (0 \underline{\leq} z \underline{\leq} z_{sat}), \frac{\partial w_s(z)}{\partial t} \underline{\equiv} \frac{\partial F_s(z)}{\partial z} + S_s(z) \quad (z_{sat} < z \underline{\leq} z_{max}), \begin{cases} \frac{\partial F_s(z)}{\partial z} - S_{\text{s}}(z) + F_{\text{c}} & (0 \leq z < z_{\text{t}}), \\ \frac{\partial F_s(z)}{\partial z} - S_{\text{s}}(z) & (z_{\text{t}} < z \leq z_{\text{b}}), \end{cases} \quad (53)$$

where  $F_{\text{s}}(z)$ and $S_{\text{s}}(z)$ are water flux and absorption by roots at a soil depth of $z$, respectively.  $F_{\text{c}}$ is water flux from the canopy layer (Eq. 46). In the case of flooded surface, the topsoil layer is assumed to be saturated  as follows,

$$\quad \underline{w_{\text{s}}(z)} \underline{\doteq} \underline{w_{\text{sat}}} \quad \underline{(\text{if flooded}; 0 \leq z < z_{\text{t}}).} \quad (54)$$

This assumption is not considered in the original MATSIRO.  $z_{\text{t}}$ is a simulation setting parameter.  $F_{\text{s}}(z)$ is calculated from the gradient of water potentials as follows.

$$F_{\underline{s}\text{s}}(z) = \begin{cases} -K(z)\left(\frac{\partial \psi(z)}{\partial z} + 1\right) & (0 \leq z \leq z_{\text{b}}) \\ (w_{\text{sat}}/\tau_{\text{b}})(w_{\text{s}}(z)/w_{\text{sat}})^2 & (z_{\text{b}} < z \leq z_{\text{max}}) \end{cases}, \quad (55)$$

where $K(z)$ is the hydraulic conductivity and $\psi(z)$ is the water potential at a soil depth of $z$.  $F_{\text{s}}(z)$ in the bottommost layer ( $z_{\text{b}} < z \leq z_{\text{max}}$) represents the base flow, and  $\tau_{\text{b}}$ is the recession constant for base flow. This model uses a simple model for simulating base flow developed by Hanasaki et al. (2008), although the original MATSIRO utilizes a more complicated model (TOPMODEL: Beven and Kirkby (1979)).  $z_{\text{b}}$ is a simulation setting parameter, and  $\tau_{\text{b}}$ is determined as described in Hanasaki et al. (2008). $K(z)$ and $\psi(z)$ are given by Clapp and Hornberger (1978) as follows.

$$K(z) = K_{\underline{s}\text{s}}\left(\frac{\overline{w_s(z)}\, w_{\text{s}}(z)}{\overline{w_{sat}}\, w_{\text{sat}}}\right)^{2B+3}, \quad (56)$$

$$\quad \psi(z) = \psi_{\underline{s}\text{s}}\left(\frac{\overline{w_s(z)}\, w_{\text{s}}(z)}{\overline{w_{sat}}\, w_{\text{sat}}}\right)^{-B}, \quad (57)$$

where  $K_{\text{s}}$ and $\psi_{\text{s}}$ are hydraulic conductivity and water potentials at saturation, respectively, and $B$ is a parameter that determines the relationship of hydraulic conductivity or water potentials between saturated and unsaturated soils.  $K_{\text{s}}, \psi_{\text{s}}$, and $B$ are soil-type specific parameters.  $S_{\text{s}}(z)$ in Eq.  53 is calculated from the transpiration

$$S_{\underline{s}\text{s}}(z) = \begin{cases} (E_{\text{t}}/\rho_{\text{w}})f_{\text{r}}(z) & (0 \leq z \leq z_{\text{rt}}) \\ 0 & (z_{\text{rt}} < z \leq z_{\text{max}}) \end{cases}, \quad (58)$$

where $E_t$ $E_t$ is the transpiration calculated in Eq. 8and $z_{rt}$, $z_{rt}$ is a root depth calculated by the CGM (Eq. 140). In Eq. 58, $f_r(z)$ is the distribution of root and is given by

$$f_r(z) \quad = \quad (3/2)(z_{rt}^2 - z^2)/z_{rt}^3, \tag{59}$$

where we assumed that $S_s(z)$ has no dependency on soildepthroot has no spatial orientation and is equally distributed in soil. We note that the root depth and distribution in MATCRO changes, although those variables are fixed in the original MATSIRO. The humidity of topsoil, $h_{ms}$, used in Eq. 9 is given by

$$h_{ms} \quad = \quad \exp(\psi(0)g/(R_a T_s(0))). \tag{60}$$

In MATCRO, it is assumed that crop can use soil water beyond the wilting point with water potential of -1500kPa ($w_{wlt}$). Hence the maximum transpiration ($E_{t,max}$) is given by

$$E_{t,max} \quad = \quad \frac{\rho_w}{\delta t} \int_0^{z_{rt}} (w_s(z) - w_{wlt})dz, \tag{61}$$

where $w_{wlt}$ is a soil-type specific parameter, and $\delta t$ is the time resolution of simulations. In the case of non-flooded surface, evaporation from the surface ($E_g$) is limited by soil water in the topsoil layer ($0 \le z < z_t$) and is given by

$$E_{g,max} \quad = \quad \frac{\rho_w}{\delta t} \int_0^{z_t} (w_s(z))dz. \tag{62}$$

In the case of flooded surface, there is no limitation for $E_{g,max}$.

**4 Crop growth model**

The main purpose of the CGM is to simulate rice yield and biomass growth for each organ during a growing period. The CGM has four modules: "net carbon assimilation", "crop development", "crop growth", and "LAI, crop height, and root depth". Each module is described in detail in the following sections.

**4.1 Net carbon assimilation**

The main role of this module is to calculate net carbon assimilation ($A_n$ $A_n$) in canopy for simulating crop growth. In addition, the stomatal conductance per unit leaf area for both sides of the leave ($\overline{g_s}$ $\overline{g_s}$) is calculated for simulating roughness length (Appendix E). Although this module is based on the Big-leaf model (Sellers et al., 1992, 1996a) used in the original MATSIRO, we refined two points in the calculation according to the approach described by de Pury and Farquhar (1997) and Dai et al. (2004). The first refinement is that leaves in a canopy are divided into sunlit and shade leaves. Subsequently, $A_n$ $A_n$ per unit leaf area for each the sunlit and shade leaves are calculated. The second refinement is that $A_n$ $A_n$ for the entire canopy is calculated considering vertical distribution of nitrogen within the canopy.

$A_\mathrm{n}$ for the entire canopy is given by

$$A_\mathrm{n} = \overline{A}_{\mathrm{n,sn}} L_\mathrm{sn} + \overline{A}_{\mathrm{n,sh}} L_\mathrm{sh}, \tag{63}$$

where $\overline{A}_\mathrm{n,sn}$ and $\overline{A}_\mathrm{n,sh}$ are net carbon assimilation per unit leaf area for sunlit and shade leaves, respectively, $L_\mathrm{sn}$ and $L_\mathrm{sh}$ are LAI for sunlit and shade leaves, respectively, and overbars represent the amounts per unit leaf area. $\overline{A}_\mathrm{n,sn}$ and $\overline{A}_\mathrm{n,sh}$ are defined by the difference between gross carbon assimilation and respiration as follows:

$$\overline{A}_\mathrm{n,x} = \overline{A}_\mathrm{g,x} - \overline{R}_\mathrm{d,x}, \tag{64}$$

where $\overline{A}_\mathrm{g,x}$ and $\overline{R}_\mathrm{d,x}$ are gross carbon assimilation and respiration per unit leaf area, respectively, and the suffix $x$ indicates sn or sh. $L_\mathrm{sn}$ and $L_\mathrm{sh}$ are given as follows.

$$L_\mathrm{sn} = \int_0^L f_\mathrm{sn}(l)dl, \tag{65}$$

$$L_\mathrm{sh} = \int_0^L (1 - f_\mathrm{sn}(l))dl, \tag{66}$$

where $f_\mathrm{sn}(l)$ is the fraction of sunlit leaves at a LAI depth of $l$ and is defined as follows:

$$f_\mathrm{sn}(l) = \exp(-Fl\sec(\theta)), \tag{67}$$

where $F$ denotes distribution of leaf orientation and $\theta$ is a zenith angle of the sun (Appendix B). The effect of photosynthesis down-regulation due to acclimatization to elevated $CO_2$ is represented as follows:

$$\overline{A}_\mathrm{g,x} = f_\mathrm{dwn} * \overline{A}_\mathrm{g',x}, \tag{68}$$

$$f_\mathrm{dwn} = \{1 + \gamma_\mathrm{gd}\ln(C_\mathrm{a,ppm}/C_0)\}/\{1 + \gamma_\mathrm{g}\ln(C_\mathrm{a,ppm}/C_0)\}, \tag{69}$$

where $\overline{A}_\mathrm{g',x}$ is gross carbon assimilation per unit leaf area for sunlit and shade leaves without photosynthesis down-regulation, $f_\mathrm{dwn}$ is the factor for photosynthesis down-regulation, $\gamma_\mathrm{gd}$ and $\gamma_\mathrm{g}$ are parameters that characterize the response to increased $CO_2$, $C_\mathrm{a,ppm}$ is atmospheric $CO_2$ concentration, and $C_0$ is the base concentration of $CO_2$. The Eqs. 68 and 69 are based on Arora et al. (2009), although the original MATSIRO does not consider the effect of photosynthesis down-regulation. We set $\gamma_\mathrm{gd} = 0.42$, $\gamma_\mathrm{g} = 0.9$, and $C_0 = 288$ according to Arora et al. (2009). It should be noted that we have tentatively set these values for the parameters of photosynthesis down-regulation, using the mean values in Arora et al. (2009), because these values are not available for rice. If these values are quantified, they should be replaced. The calculation for $\overline{A}_\mathrm{g',x}$ and $\overline{R}_\mathrm{d,x}$ is based on the leaf photosynthesis model developed by Collatz et al. (1991). In their model, $\overline{A}_\mathrm{g',x}$ is determined by three limiting factors: Rubisco, light, and sucrose synthesis, as follows:

$$\overline{A}_\mathrm{g',x} \le \min(\overline{\omega}_\mathrm{c,x}, \overline{\omega}_\mathrm{e,x}, \overline{\omega}_\mathrm{s,x}), \tag{70}$$

where $\overline{\omega}_{c,x}, \overline{\omega}_{e,x},$ and $\overline{\omega}_{s,x}$ are Rubisco-limited, light-limited, and sucrose-limited carbon assimilation per unit leaf area, respectively. To implement smooth transition between each limited state, $\overline{A}_{g',x}$ is determined practically by solving the following two equations (Sellers et al., 1996b):

$$\beta_{ce}\overline{\omega}^2_{p,x} - \overline{\omega}^2_{p,x}(\overline{\omega}^2_{c,x}+\overline{\omega}^2_{e,x})+\overline{\omega}^2_{c,x}\overline{\omega}^2_{e,x} = 0 \tag{71}$$

$$\beta_{ps}\overline{A}^2_{g',x} - \overline{A}^2_{g',x}(\overline{\omega}^2_{p,x}+\overline{\omega}^2_{s,x})+\overline{\omega}^2_{p,x}\overline{\omega}^2_{s,x} = 0, \tag{72}$$

where $\beta_{ce}$ and $\beta_{pc}$ are the parameters that determine the smoothness of transition between each limited state. $\beta_{ce}$ is a crop-specific parameter and $\beta_{pc}$ is a parameter that does not depend on crop type. The variables $\overline{\omega}_{c,x}, \overline{\omega}_{e,x},$ and $\overline{\omega}_{s,x}$ are given by

$$\overline{\omega}_{c,x} = \overline{V}_{mc,x}\left\{\frac{c_{i,x}-\Gamma^*}{c_{i,x}+K_c(1+[O_2]/K_O)}\right\} \tag{73}$$

$$\overline{\omega}_{e,x} = \epsilon_e\overline{Q}_x\left\{\frac{c_{i,x}+\Gamma^*}{c_{i,x}+2\Gamma^*}\right\} \tag{74}$$

$$\overline{\omega}_{s,x} = \overline{V}_{ms,x}/2. \tag{75}$$

Here, $\overline{V}_{mc,x}$ and $\overline{V}_{ms,x}$ are the maximum Rubisco capacity per unit leaf area for $\overline{\omega}_{c,x}$ and $\overline{\omega}_{s,x}$, respectively, $c_{i,x}$ is the partial pressure of intercellular $CO_2$, $[O_2]$ is the partial pressure of intercellular $O_2$, $\overline{Q}_x$ is the photon flux density for PAR absorbed per unit leaf area by sunlit and shade leaves, $\epsilon_e$ is the quantum efficiency, $\Gamma^*$ is the light compensation point, and $K_c$ and $K_O$ are the Michaelis constant for $CO_2$ fixation and oxygen inhibition, respectively. We set $[O_2] = 20{,}900$ (Collatz et al., 1991). $\epsilon_e$ is a crop specific parameter. $\overline{V}_{mc,x}$ and $\overline{V}_{ms,x}$ are given by

$$\overline{V}_{mc,x} = \overline{V}_{max,x}2^{q_t}/\{1+\exp\left(s_1(T_c - s_2)\right)\}f_v, \tag{76}$$

$$\overline{V}_{ms,x} = \overline{V}_{max,x}2^{q_t}/\{1+\exp\left(s_3(s_4 - T_c)\right)\}f_v, \tag{77}$$

where $\overline{V}_{max,x}$ is the reference value for the maximum Rubisco capacity per unit leaf area of sunlit ($\overline{V}_{max,sn}$) and shade ($\overline{V}_{max,sh}$) leaves, $f_v$ is the water stress factor, $s_1$, $s_2$, $s_3$, and $s_4$ are parameters that represent temperature dependence of $\overline{V}_{max,x}$ on $\overline{V}_{mc,x}$ or $\overline{V}_{ms,x}$, $q_t$ is a function that represents temperature dependency. The variables $s_1$ and $s_2$ are parameterised in Masutomi et al. (2016), whereas $s_3$ is a parameter that does not depend on crop type and $s_4$ is a crop-specific parameter. $f_v$ is given by

$$f_v = \int_0^{r_t} f_r(T_c - 298z)/10 \cdot f_s(z)dz, \tag{78}$$

$$f_s(z) = \frac{2}{1+\exp(-\gamma_s\psi_s(z))}, \tag{79}$$

[revised manuscript text omitted]

$$
10 \quad h_{s,x}\text{s},x \;=\; e_{s,x}\text{s},x / e_{sat}\text{sat}(T_c\text{c}, P_a\text{a}), \tag{102}
$$

where  $e_{s,x}$ is the vapour pressure at leaf boundary and  $e_{sat}$ is the saturated vapour pressure. The variable  $e_{s,x}$ is expressed as

$$
e_{s,x}\text{s},x \;=\; (e_a\text{a}\,\overline{g}_l\text{l} + e_i\text{i}\,\overline{g}_{st,x}\text{st},x)/(\overline{g}_l\text{l} + \overline{g}_{st,x}\text{st},x), \tag{103}
$$

where  $e_a$ and $e_i$ are the vapour pressure in the air and leaf, respectively. Eq. 103 is derived from the fact that the water
15   vapour flux from the stomata to leaf surface is equal to the water vapour flux from the leaf surface into the atmosphere, which is shown in the following equation:

$$
\overline{g}_{st,x}\text{st},x(e_i\text{i} - e_s) \;=\; \overline{g}_l\text{l}(e_{s,x}\text{s},x - e_a\text{a}). \tag{104}
$$

The parameters  $e_a$, $e_i$, and $e_{sat}$ are given by

$$
e_a\text{a} \;=\; Q(R_{dry}\text{dry}/R_{vap}\text{vap}), \tag{105}
$$

$$
20 \quad e_i\text{i} \;=\; e_{sat}\text{sat}(T_c\text{c}, P_a\text{a}), \tag{106}
$$

$$
e_{sat}\text{sat}(T_c\text{c}, P_a\text{a}) \;=\; Q_{sat}\text{sat}(T_c\text{c}, P_a\text{a})(R_{dry}\text{dry}/R_{vap}\text{vap}), \tag{107}
$$

where  $e_i$ is assumed to be saturated.

Now we have three relationships (Eqs. 64 to 94, Eq. 96, and Eq. 101) in terms of three unknown variables ( $\overline{A}_{n,x}, c_{i,x}$, and $\overline{g}_{st,x}$). Therefore, we can determine the values for  $\overline{A}_{n,x}, c_{i,x}$, and $\overline{g}_{st,x}$, by numerically
25   solving the three relationships. The numerical method is described in Masutomi et al. (2016).

Last,  $\overline{g}_s$ is given by the following equation:

$$
\overline{g}_s\text{s} \;=\; \overline{g}_{st}\text{st} * (T_c\text{c}\,R_{vap}\text{vap}\,w_{H_2O}\text{H}_2\text{O}/P_a\text{a}), \tag{108}
$$

$$
\overline{g}_{st}\text{st} \;=\; \{(\overline{g}_{st,sn}\text{st},sn * L_{sn}\text{sn} + \overline{g}_{st,sh}\text{st},sh * L_{sh}\text{
[revised manuscript text omitted]
\tag{132}
$$

where $D_{\mathrm{vs,rot1}}, D_{\mathrm{vs,rot2}}, D_{\mathrm{vs,lef1}}, D_{\mathrm{vs,lef2}}, D_{\mathrm{vs,pnc1}},$ and $D_{\mathrm{vs,pnc2}}$ represent the $D_{\mathrm{vs}}$ values at which corresponding partitions change, $P_{\mathrm{rot}}$ is the ratio of partitioned glucose to the roots at $D_{\mathrm{vs}} \le D_{\mathrm{vs,rot1}}$, and $P_{\mathrm{lef}}$ is the ratio of glucose partitioned to the leaf and glucose partitioned to shoot at $D_{\mathrm{vs}} \le D_{\mathrm{vs,lef1}}$. $D_{\mathrm{vs,rot1}}, D_{\mathrm{vs,rot2}}, D_{\mathrm{vs,lef1}}, D_{\mathrm{vs,lef2}}, D_{\mathrm{vs,pnc1}}, D_{\mathrm{vs,pnc2}}, P_{\mathrm{rot}},$ and $P_{\mathrm{lef}}$ are crop-specific parameters and are parameterized in Masutomi et al. (2016). In Eq. 130, we assume that no glucose is partitioned to shoot during transplanting shock ($D_{\mathrm{vs,tr}} < D_{\mathrm{vs}} \le D_{\mathrm{vs,te}}$). It is important to note that transplanting shock is considered only when transplanting is conducted.

Loss of leaf dry weight due to leaf death ($L_{\mathrm{s,lef}}$) and remobilization from starch reserve in stem ($R_{\mathrm{m,stm}}$) occur after heading and they are defined as follows

$$
L_{\mathrm{s,lef}} = \begin{cases} 0 & (D_{\mathrm{vs}} \le D_{\mathrm{vs,h}}), \\ r_{\mathrm{dd,lef}}(W_{\mathrm{lef}} + W_{\mathrm{glu}}) & (\text{Otherwise}) \end{cases}
\tag{133}
$$

$$
R_{\mathrm{m,stc}} = \begin{cases} 0 & (D_{\mathrm{vs}} \le D_{\mathrm{vs,h}}), \\ r_{\mathrm{rm,stc}}W_{\mathrm{stc}} & (\text{Otherwise}) \end{cases}
\tag{134}
$$

where $r_{\mathrm{dd,lef}}$ and $r_{\mathrm{rm,stc}}$ represent the ratios of leaf death and remobilization. $r_{\mathrm{dd,lef}}$ varies with $D_{\mathrm{vs}}$ as follow:

$$
r_{\mathrm{dd,lef}} = r_{\mathrm{d1,lef}}(D_{\mathrm{vs}} - D_{\mathrm{vs,h}})/(1 - D_{\mathrm{vs,h}})
\tag{135}
$$

where $r_{\mathrm{d1,lef}}$ is the ratio of leaf death at harvest ($D_{\mathrm{vs}} = 1$) and it is parameterized in Masutomi et al. (2016). We set $r_{\mathrm{rm,stc}} = 1.16 * 10^{-6}$, assuming that all starch stored in stem is remobilized in 10 days after heading (Bouman et al., 2001).

Last, the dry weight of shoot ($W_{\mathrm{sh}}$), used in Section 3.4, is given by

$$
W_{\mathrm{sh}} = W_{\mathrm{lef}} + W_{\mathrm{stm}} + W_{\mathrm{pnc}} + W_{\mathrm{stc}} + W_{\mathrm{glu}}.
\tag{136}
$$

**4.4 LAI, crop height, and root depth**

Leaf area index ($L$), crop height ($h_{\mathrm{gt}}$), and root depth ($z_{\mathrm{rt}}$) are expressed as

$$
L = (W_{\mathrm{lef}} + W_{\mathrm{glu}})/S_{\mathrm{lw}},
\tag{137}
$$

$$
S_{\mathrm{lw}} = S_{\mathrm{lw,mx}} + (S_{\mathrm{lw,mn}} - S_{\mathrm{lw,mx}})\exp(-k_{SLW}D_{\mathrm{
[revised manuscript text omitted]

$$A^+ = \frac{c_m L}{2\kappa^2}\frac{c_{\text{m}} L}{2\kappa^2},$$

$$C^0{}_{XX} = \left(\ln\frac{h_{gt}-d}{z_M}\ln\frac{h_{\text{gt}}-d}{z_{\text{M}}}\right)^{-1}\left(\ln\frac{h_{gt}-d}{z_X^+}\ln\frac{h_{\text{gt}}-d}{z_X^+}\right)^{-1},$$

$$C^\infty{}_{XX} = \frac{-1+(1+8F_X)^{0.5}}{2}\frac{-1+(1+8F_X)^{0.5}}{2},$$

$$F_{XX} = \frac{c_X}{c_m}\frac{c_X}{c_{\text{m}}},$$

$$\left(\ln\frac{h_{gt}-d}{z_*^+}\ln\frac{h_{\text{gt}}-d}{z_*^+}\right)^{-1} = \frac{1}{-\ln\left(\frac{z_{*s}}{h_{gt}}\right)}\frac{1}{-\ln\left(\frac{z_{*s}}{h_{\text{gt}}}\right)}\left(\frac{P_{1*}}{P_{1*}+A^+\exp(A^+)}\right)^{P_{2*}},$$

$$P_{1*} = 0.00115\left(\frac{z_{*s}}{h_{gt}}\frac{z_{*s}}{h_{\text{gt}}}\right)^{0.1}\exp\left\{5\left(\frac{z_{*s}}{h_{gt}}\right)\right\}\exp\left\{5\left(\frac{z_{*s}}{h_{\text{gt}}}\right)\right\},$$

$$P_{2*} = 0.55\exp\left\{-0.58\left(\frac{z_{*s}}{h_{gt}}\right)^{0.35}\right\}\exp\left\{-0.58\left(\frac{z_{*s}}{h_{\text{gt}}}\right)^{0.35}\right\},$$

$$P_{3X} = \{F_{XX} + 0.084\exp(-15F_X)\exp(-15F_X)\}^{0.15},$$

$$P_{4X} = 2F_X^{1.1},$$

$$c_{ee} = c_{hh}/(1 + c_{hh}(U_{cc}/\overline{g}_{ss})).$$

[revised manuscript text omitted]